# Schemas provide a scaffold for neocortical integration of new memories over time

**Sam Audrain** [1,2] ✉ **& Mary Pat McAndrews** [1,2]

Memory transformation is increasingly acknowledged in theoretical accounts of systems consolidation, yet how memory quality and neural representation change over time and how schemas influence this process remains unclear. We examined the behavioral quality and neural representation of schema-congruent and incongruent object-scene pairs retrieved across 10-minutes and 72-hours using fMRI. When a congruent schema was available, memory became coarser over time, aided by post-encoding coupling between the anterior hippocampus and medial prefrontal cortex (mPFC). Only schema-congruent representations were integrated in the mPFC over time, and were organized according to schematic context. In the hippocampus, pattern similarity changed across 72-hours such that the posterior hippocampus represented specific details and the anterior hippocampus represented the general context of specific memories, irrespective of congruency. Our findings suggest schemas are used as a scaffold to facilitate neocortical integration of congruent information, and illustrate evolution in hippocampal organization of detailed contextual memory over time.

As we go about our lives, we are constantly experiencing new events and details that are unique to a given moment and place in time. As only a small subset of our experiences are remembered long-term, what neural underpinnings support the retention of certain events and the loss of others? We know that successful long-term memory retrieval initially depends on the hippocampus and gradually comes to be supported by the neocortex over time through the process of systems consolidation. Studies of animal and human memory systems indicate that recall of detail-rich episodic memories remain dependent on the hippocampus in perpetuity, while hippocampal-neocortical dialog promotes the strengthening of neocortical representation such that coarse, schematic memories can be supported by the neocortex independent of the hippocampus with time[1–9], c.f. refs. [10],[11]. However, an important consideration was not initially appreciated in models of systems consolidation, namely, that memories are not written to a blank neocortical slate. As adults, we have years of experience interacting with the world, with many experiences represented in the brain in distributed neural ensembles as long-term memories and schematic knowledge. How does what we already know influence what we will

remember of a new experience? How do our prior experiences influence mental and neural representation over time?

In recent years there is increased understanding of how prior knowledge, and schemas that are extracted from multiple similar experiences in particular, can enhance memory acquisition, consolidation, and retrieval[12–16]. While the establishment of the neocortical memory trace was originally conceived of as a slow process unfolding over weeks to years[17,18], in rodents, there is evidence that learning novel information in the context of an existing schema occurs quite quickly, effectively accelerating consolidation such that new information can be retrieved without the hippocampus faster than usual[19]. While schema-accelerated consolidation has yet to be definitively demonstrated in humans, it has long been recognized that prior knowledge benefits mnemonic retention of new congruent information[20–22].

Empirically, the medial prefrontal cortex (mPFC; and homologous regions in rodents) has proven to be important for schema benefits to memory across both species[9,14,23,24]. There is evidence that hippocampal activity decreases[25–28] while mPFC activity increases during the delayed retrieval of schema-congruent relative to incongruent

[1]Division of Clinical and Computational Neuroscience, Krembil Research Institute, University Health Network, Toronto, ON M5T 2S8, Canada. [2]Department of Psychology, University of Toronto, Toronto, ON M5S 3G3, Canada. ✉e-mail: samantha.audrain@mail.utoronto.ca

memories[26,29,30], indicating the increasing contribution of the mPFC in supporting schema-congruent retrieval after a period of consolidation. Further, functional coupling between the mPFC and hippocampus increases during encoding of information related to prior knowledge, and persists off-line after learning[26,31–34], c.f. refs. 35,36, which is proposed to reflect updating of neocortical knowledge structures with related experiences[37,38].

Although a relative trade off in activity between the mPFC and hippocampus provides some support for the existence of schema-accelerated consolidation in humans, it is unclear exactly how this might occur. Recent theoretical accounts propose that schemas provide an organizing scaffold that new overlapping content can leverage to facilitate integration[14,39]. Such accounts rest on the idea that related memories are represented by overlapping neural ensembles in the neocortex−and the overlap of new with established content enables the rapid strengthening of new representations via Hebbian learning[15], presumably at the cost of memory specificity afforded by the hippocampus. The implication is that schema-congruent memories are organized and integrated according to the schema that supports acquisition, which can then be mobilized to facilitate later encoding and retrieval[14].

There is indeed evidence that learned arbitrary associations that share overlapping features come to be represented more similarly to each other in the mPFC than nonoverlapping events[40,41], lending credence to the contention that overlapping information is strengthened in this region while details fall away leaving coarser, more integrated representations[39,41]. Although paradigms measuring integration as a function of shared arbitrary features can speak to the role of overlap in linking episodic memories, we argue that the complex and abstracted real-world knowledge that comprise schemas are likely more efficacious for integration of new information, as they are not limited to the arbitrary elements that are presented in an experimental context. Neural population overlap as a mechanism for integration has not been examined in the context of schemas, leaving untested the idea that representational overlap facilitates neocortical integration. Moreover, while memory transformation and quality of memory are becoming increasingly acknowledged in theoretical accounts of long-term memory formation[2,4,13,38,39,42–44], there is a dearth of work empirically examining how memory quality and representation change over time, and how schemas affect this process. Using resting-state fMRI, neural pattern similarity analyses, and a novel behavioral paradigm sensitive to quality of memory, we targeted several lines of evidence to address the following questions: do real-world schemas act as a scaffold to enhance neocortical integration of new overlapping memories in humans, and how do the hippocampus and mPFC interact to support the consolidation and retrieval of coarse and detailed episodic representations in the context of schemas over time?

With these aims, participants studied a series of unique objects that were congruent or incongruent with background scenes in an event-related fMRI paradigm (Fig. 1). During retrieval, they were presented with the object cue and were asked to retrieve the background, indicating both the context that the object had been paired with (kitchen or beach), and the specific scene (which specific kitchen or beach). Memory was tested across a short delay of 10 min and a long delay of 3 days, to measure change in memory specificity and neural representation over time. We used a multi-voxel pattern analysis approach to quantify the degree of overlap in memory representations as well as baseline and post-encoding resting state scans to measure experience dependent changes in hippocampal-neocortical interaction as it relates to quality of subsequent memory. We investigated three hypotheses that speak to the facilitated integration of schema-congruent information and its organization: (1) Schema-congruent memories should become coarser than incongruent ones over time. (2) Stronger post-encoding hippocampal-mPFC coupling should associate with coarser memory for schema-congruent information

over time. Finally, (3) over time, representations of schema-congruent memories should become integrated according to the schematic context with which they are related, and thus overlap in the mPFC, whereas hippocampal representations could remain distinct when individuals are able to retrieve fine details. In sum, we targeted three potential converging lines of evidence that together could substantiate the phenomenon of schema-facilitated neocortical integration of overlapping information.

Here, we show that memory for schema-congruent associations becomes coarser over time, which is associated with post-encoding coupling between the anterior hippocampus and mPFC. We also show that schema-congruent representations are integrated in the mPFC over time, and are organized according to the schematic context available during learning. Finally, we show that pattern similarity in the posterior hippocampus comes to represent specific details over time, while the anterior hippocampus comes to represent the general context of specific memories, irrespective of congruency. The present study provides evidence that schemas are used as a scaffold to facilitate neocortical integration of related information, and suggests that hippocampal organization of detailed contextual memory evolves over time.

## Results

### Congruent associations are remembered more coarsely over time

To examine the influence of schema congruency on memory over time, total percent correct was calculated for each participant as the percent of congruent or incongruent pairs in which the correct context was retrieved (regardless of the specific scene) at each delay. The total percent correct score is therefore comprised of both coarse and detailed memories. We submitted total percent correct scores to a linear mixed effects model with congruency (congruent/incongruent) and delay (short/long) as fixed effects, a random intercept for each participant, and random slopes for counterbalancing groups. The model indicated main effects of congruency ($F(1,58) = 72.34$, $p < 0.0001$, M difference $= -32.43$, CI: $[-40.07-(-24.78)]$) and delay ($F(1,58) = 201.07$, $p < 0.0001$, M difference $= 11.52$, CI: $[5.91-17.12]$), and a congruency by delay interaction ($F(1,58) = 19.45$, $p < 0.001$, M difference $= 20.56$, CI: $[11.23-29.89]$). Pairwise testing revealed that congruent pairs were remembered better than incongruent ones at the short delay ($t(58) = 4.43$, $p < 0.0001$, M difference $= 11.86$, CI: $[6.51-17.22]$). While both types of pairs where forgotten over time (congruent: $t(58) = 4.11$, $p = 0.0001$, M difference $= -11.52$, CI: $[-17.12-(-5.91)]$, incongruent: $t(58) = 8.58$, $p < 0.0001$, M difference $= -32.08$, CI: $[-39.56-(-24.59)]$), congruent pairs were retained better at the long delay compared to incongruent ($t(58) = 8.49$, $p < 0.0001$, M difference $= 32.43$, CI: $[24.78-40.07]$).

Next, we examined how quality of memory changed over time within each condition (Fig. 2). At each delay, we defined detailed memories as the percent of total congruent or incongruent trials where participants correctly identified both the context (beach/kitchen) and the specific scene that was paired with a given object at retrieval. We defined coarse memories as the percent of total congruent and incongruent trials in which participants correctly identified the context an object had been paired with, but indicated that they did not know the specific scene, or chose the incorrect scene of the same context. We present the proportion of trials for each condition for which participants indicated "don't know" versus chose the incorrect context or scene in Supplementary Method 1/Supplementary Fig. 1. We ran separate linear mixed models for coarse and detailed memories, with congruency and delay as predictors, with a random intercept for each participant, and random slopes for each counterbalancing group.

For detailed memories, we found a significant main effect of delay ($F(1,58.87) = 143.20$, $p < 0.0001$, M difference $= 25.94$, CI: $[18.88-33.76]$) as memories were forgotten over time. There was

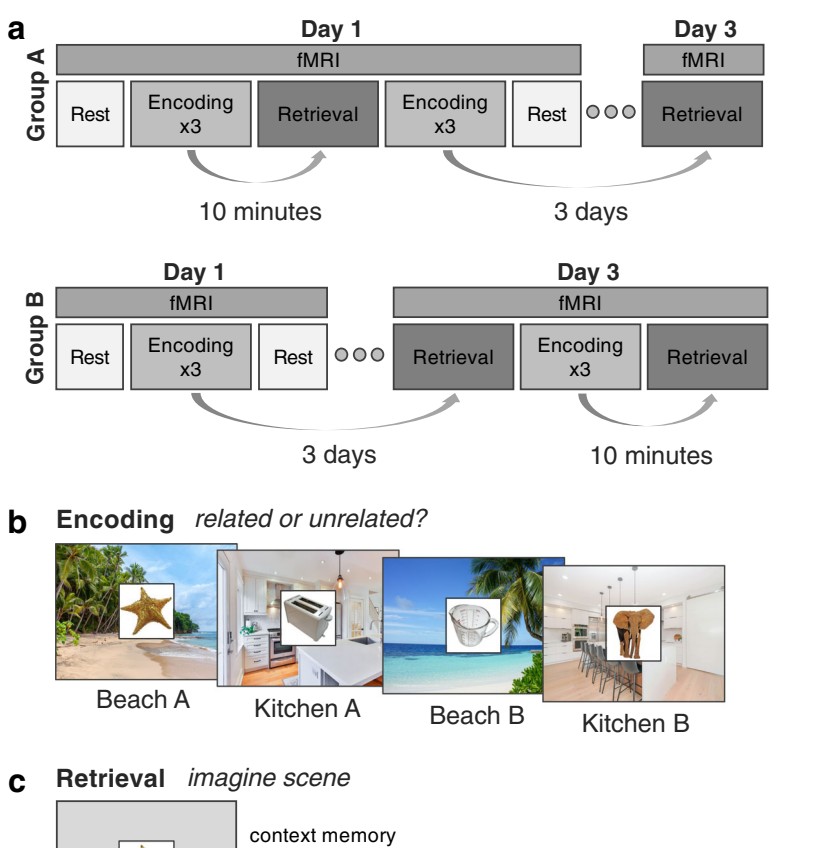

**Fig. 1 | Experimental design. a** Participants underwent encoding and retrieval sessions across a short delay of 10 min and a long delay of 3 days in the fMRI scanner. Participants were counterbalanced across delays according to the depicted structure. Encoding fMRI data was not analyzed for the present experiment. **b** During encoding participants viewed a series of objects, each paired with one of four repeating scenes (two beaches/two kitchens) and indicated whether the object was related to the background scene. Participants did not view the names of the scenes during encoding, rather, they learned the names of the scenes and practiced visualizing them in detail prior to beginning the experiment. **c** During retrieval participants were asked to imagine the scene associated with a presented object in as much detail as possible, and to indicate which context the object had been paired with (kitchen or beach), as well as which specific scene (which kitchen or which beach). Trials for which participants remembered the context a given object was paired with but not the scene were scored as coarse memories, and trials for which they remembered the context and specific scene were scored as detailed memories. Dark gray circles over responses here represent example responses. Scene and object images presented here are placeholders used for illustrative purposes. Objects were retrieved from the bank of standardized stimuli (BOSS) database (Copyright (C) 2009, 2010 Mathieu Brodeur)[100]. Beach A photo by Rowan Heuvel, Beach B photo by Pedro Monteiro, Kitchen A photo by Sidekix Media, Kitchen B photo by Zac Gudakov, all on Unsplash: https://unsplash.com/license.

also a main effect of congruency ($F(1,57.17) = 36.04$, $p < 0.0001$, M difference = −19.43, CI:[−26.29−(−12.40)]), as detailed congruent pairs were better remembered than incongruent. The interaction between delay and congruency for detailed memories was marginal ($F(1,57.17) = 3.58$, $p = 0.063$, M difference = 9.32, CI: [−0.45−19.02]). For coarse memories, there were significant main effects of delay ($F(1,62.76) = 25.05$, $p < 0.0001$, M difference = −1.66, CI: [−2.34−(−1.04)]) and congruency ($F(1,58.43) = 12.15$, $p = 0.0009$, M difference = −1.29, CI: [−1.92−(−0.58)]), and a significant interaction between the two ($F(1,58.43) = 5.15$, $p = 0.027$, M difference = 1.02, CI: [0.089−1.91]). Pairwise $t$-tests indicated that there was no difference in the percentage of coarse congruent and incongruent memories at the short delay ($t(58.4) = 0.90$, $p = 0.37$, M difference = 0.27, CI: [−0.33−0.88]). There were, however, more coarse congruent than incongruent memories retrieved at the long delay ($t(58.4) = 3.89$, $p = 0.0003$, M difference = 1.29, CI: [0.63−1.96]), which was driven

by an increase in the percentage of coarse congruent memories retrieved over time ($t(60.7) = 5.16$, $p < 0.0001$, M difference = 1.66, CI: [1.02−2.30]). There was a smaller, marginal increase in the percentage of coarse incongruent memories over the same time period ($t(60.7) = 1.99$, $p = 0.051$, M difference = 0.64, CI: [−0.004−1.28]). The finding that memory became coarser over time in the congruent but not incongruent condition is consistent with the hypothesis that congruency with prior knowledge facilitates neocortical integration of new memories beyond that which occurs for incongruent associations across the 3 days. Subsequent control analyses confirmed that the increase in coarse congruent memory over time was not due to differing numbers of coarse and detailed memories retained across the short delay (Supplementary Method 2; Supplementary Fig. 2), nor due to a possible increase in bias to choose the congruent context over time (Supplementary Method 3; Supplementary Fig. 3).

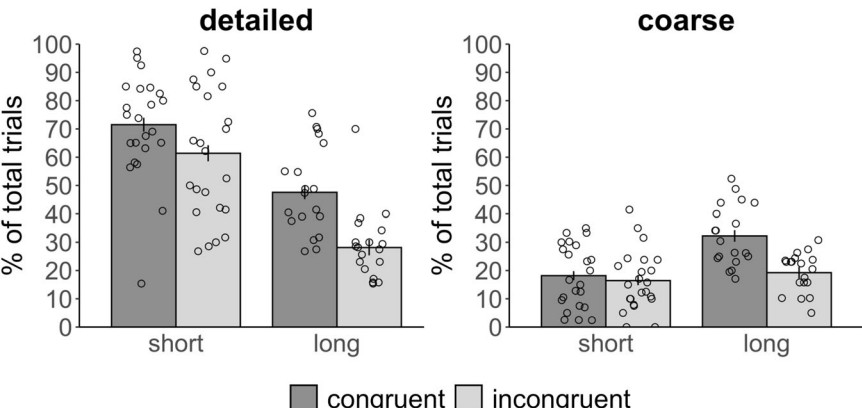

**Fig. 2 | Quality of memory for congruent and incongruent associations over time.** We examined the percent of total congruent and incongruent trials that were correctly retrieved as a function of memory granularity. Memories were considered coarse if participants retrieved the correct context an object had been paired with but not the specific scene and were considered detailed if they retrieved the specific scene. Coarse and detailed memories together comprise total memory accuracy. Error bars reflect standard error of the mean adjusted for within-subject design. $N = 23$ participants across the short delay and $N = 19$ participants across the long delay. Source data are provided as a Source Data file.

## Hippocampus – mPFC post-encoding coupling and subsequent memory

Prior research suggests that post-encoding connectivity in memory-relevant networks reflects early systems consolidation processes, with enhanced delayed long-term retrieval associated with increased connectivity[45–47]. The anterior hippocampus has greater structural and functional connectivity to the mPFC than the posterior hippocampus[42,48–50], and coupling between the anterior hippocampus and mPFC has proven to be related to mnemonic representation and retrieval of remote memories[41,51]. In order to determine if post-encoding anterior hippocampal-mPFC coupling was associated with coarse congruent memories over time, we extracted functional connectivity between the anterior hippocampus and the mPFC at baseline and after encoding object-scene pairs for the long delay. We then subtracted each participant's baseline from their post-encoding connectivity to acquire a measure of change in connectivity after encoding, which we correlated with behavioral retrieval scores across the long delay. In line with our hypothesis, we found that participants with greater post-encoding change in coupling between the anterior hippocampus and mPFC retrieved coarser congruent memories 3 days later ($t(15) = 1.98$, $p = 0.033$, $r = 0.46$, CI: [0.05–1]; Fig. 3).

If the relationship between connectivity and memory was not reflective of consolidation processes and instead indexed memory granularity regardless of having just undergone encoding, one might expect that it wouldn't matter if the connectivity data were collected right after encoding or at some arbitrary timepoint. To test if the relationship between connectivity and coarse congruent memory was evident regardless of when the data were collected, we additionally correlated change in connectivity after encoding for the long delay, with coarse congruent memory scores across the short delay. Connectivity did not reliably correlate with coarse congruent memory across the short delay ($t(15) = 0.11$, $p = 0.92$, $r = −0.03$, CI: [−0.5–0.46]), which is in line with the notion that post-encoding interaction between these regions reflects processes associated with memory consolidation rather than memory granularity per se. Exploratory analyses indicated that connectivity between the anterior hippocampus and mPFC did not reliably associate with the other memory conditions across the long delay, although the magnitudes of some were within the range of the coarse congruent condition (detailed congruent memory: $t(15) = 0.96$, $p = 0.35$, $r = −0.24$, CI: [−0.65-0.27]; coarse incongruent memory: $t(15) = 0.77$, $p = 0.46$, $r = 0.19$, CI: [−0.32–0.62]; detailed incongruent memory: $t(15) = 1.70$, $p = 0.11$, $r = −0.40$, CI:

[−0.74–0.10]; see Supplementary Method 4 and Supplementary Fig. 4 for a comparison of slopes).

## Memories are integrated in the mPFC according to congruent context

To test the hypothesis that representations of schema-congruent memories become integrated in the mPFC over time, we used a multi-voxel pattern analysis approach[41,52] to quantify the degree to which representations of schema congruent and incongruent memories overlap in the mPFC at each delay. We reasoned that higher pattern similarity between trials within a condition would reflect commonalities in neural representation. An increase in representational similarity over time, therefore, is consistent with the idea that there is increased neural population overlap and integration of mnemonic representations after a period of consolidation.

For congruent pairs, we extracted the pattern of voxels within the mPFC as participants were viewing a given object and successfully retrieving the associated context (regardless of if the specific scene was retrieved), and then correlated the extracted pattern with all other patterns for congruent object-context pairs that shared the same context (within context correlations), and with those from congruent object-context pairs of the opposing context (across context correlations; Fig. 4a). For incongruent pairs, we computed the same set of correlations except all object-context pairs were incongruent. We included both coarse and detailed trials in order to increase statistical power, because presumably, coarse information should be retrieved for both trial types (e.g., general features of beaches). If commonalities across object-context pairs are enhanced with consolidation due to congruency with pre-existing associations, an increase in pattern similarity should be evident in the mPFC for congruent information relative to incongruent despite matched contextual overlap. To the extent that schemas act as a scaffold for integration, increased pattern similarity over time should be context-specific (i.e., greater pattern similarity within-context than across contexts). As we were mainly interested in change in pattern similarity in the mPFC over time as it relates to congruency and did not have a priori hypotheses regarding hippocampal patterns along this dimension, we focused on the mPFC for this analysis (Fig. 4b). We present a plot of corresponding pattern similarity in the anterior and posterior hippocampi in Supplementary Method 5/Supplementary Fig. 5 for the interested reader.

As a linear mixed model predicting pattern similarity in the mPFC as a function of congruency (congruent/incongruent), context (within/across) and delay (short/long) was too complex to converge whilst also

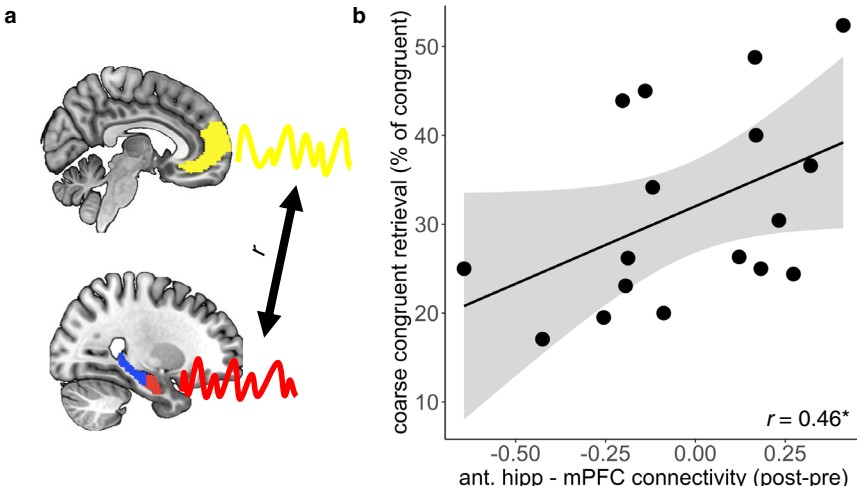

**Fig. 3 | Anterior hippocampus – mPFC increase in post-encoding coupling is associated with coarse congruent memory 3 days later. a** Time series from the anterior hippocampus and mPFC were extracted and correlated during baseline and post-encoding rest. Pre-encoding anterior hippocampus – mPFC connectivity was subtracted from post-encoding connectivity to derive a measure of post-encoding change in connectivity. **b** Correlation between anterior hippocampus and mPFC post-encoding change in coupling and coarse congruent memory 3 days later ($t(15) = 1.98$, $p = 0.033$, $r = 0.46$, CI: [0.05–1]). Coarse congruent memory was quantified as the percent of total congruent judgements for which the context was correctly retrieved but the specific scene was not. The black dots reflect data from individual participants, and the black line reflects the line of best fit for the correlation. Gray ribbon represents 95% confidence interval for a one-tailed test.*significant correlation according to a one-tailed Pearson's correlation test. Source data are provided as a Source Data file. Brains are the MNI152 template (0.5 mm, linearly smoothed; Copyright (C) 1993–2009 Louis Collins, McConnell Brain Imaging Centre, Montreal Neurological Institute, McGill University[101]), with ROI masks overlayed. The mPFC mask is a combination of A14m and A10m ROIs from the Brainnetome Atlas[92]. The anterior and posterior hippocampal masks are ahipp and chipp masks from the Brainnetome Atlas and are used for illustrative purposes, as the hippocampal masks used in this study were segmented based on each participant's anatomy. $r$ = Pearson's correlation.

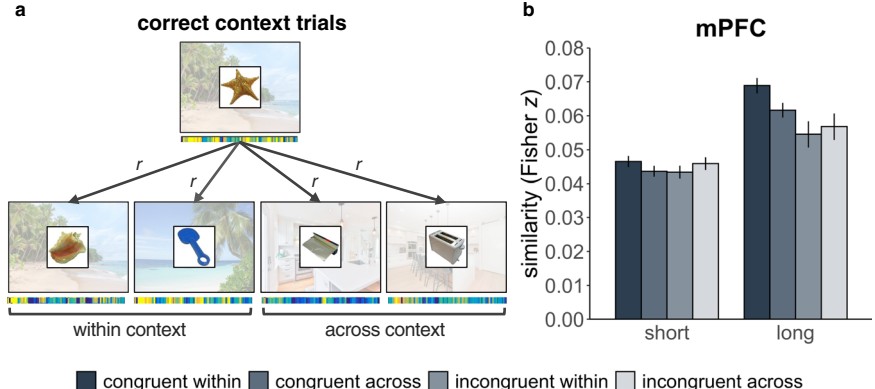

**Fig. 4 | Pattern similarity analysis in the mPFC during retrieval of congruent and incongruent object-context pairs. a** Schematic example of our analysis approach. Patterns for trials where participants successfully retrieved the context associated with the presented object (regardless of whether they retrieved the specific scene) were extracted from the mPFC and correlated within and across context, separately for congruent and incongruent trials. This example depicts congruent beach stimuli, but the same analysis was applied to incongruent beaches, as well as congruent and incongruent kitchens. Background scenes were not presented during retrieval, but were retrieved from memory. **b** Resulting pattern similarity in the mPFC over time, according to congruency (congruent/incongruent) and context (within/across). Data reflect estimated marginal means from linear mixed effects models predicting pairwise Fisher transformed correlations from congruency, context, and delay variables, in $N = 23$ participants across the short delay and $N = 19$ participants across the long delay. Errors bar reflect standard error of the mean adjusted for within-subject design. $r$ = Pearson's correlation. Scene and object images presented here are placeholders used for illustrative purposes. Objects were retrieved from the bank of standardized stimuli (BOSS) database (Copyright (C) 2009, 2010 Mathieu Brodeur)[100]. Beach photos by Rowan Heuvel and Pedro Monteiro, kitchen photos by Sidekix Media and Zac Gudakov, all on Unsplash: https://unsplash.com/license. Source data are provided as a Source Data file.

accounting for evident heteroskedasticity and participant-specific effects in the model, we ran three simpler models targeting our main questions of interest. First, we tested the hypothesis that overlapping (within-context correlations) congruent pairs become better integrated in the mPFC than overlapping incongruent ones over time (congruent within versus incongruent within bars in Fig. 4b). We ran a linear mixed effects model predicting within-context pattern similarity in the mPFC as a function of congruency and delay, with random intercepts for each participant, random slopes for each

counterbalancing group, and weighted to model unequal variance across fixed effects. We found main effects of congruency ($F(1,13848) = 17.72$, $p < 0.0001$, M difference = −0.015, CI: [−0.023–(−0.006)]) and delay ($F(1,13848) = 70.99$, $p < 0.0001$, M difference = −0.022, CI: [−0.028–(−0.017)]) as well as a significant interaction between the two ($F(1,13848) = 5.50$, $p = 0.019$, M difference = 0.011, CI: [0.002–0.021]). Pairwise $t$-tests indicated that there was no reliable difference in representational similarity of congruent and incongruent object-context pairs at the short delay

($t$(13848) = 1.37, $p$ = 0.170, M difference = 0.003, CI: [−0.001–0.008]). While patterns for both congruent and incongruent pairs became more similar within congruency over time (congruent: $t$(13848) = 8.35, $p$ < 0.0001, M difference = 0.022, CI: [0.017–0.028]; incongruent: $t$(13848) = 2.72, $p$ = 0.007, M difference = 0.011, CI: [0.003–0.019]), this change was greater for the congruent pairs such that there was greater pattern similarity for congruent than incongruent pairs at the long delay ($t$(13848) = 3.46, $p$ = 0.0006, M difference = 0.015, CI: [0.006–0.023]). In other words, object-context representations became more similar to each other in the mPFC over time if they shared a congruent context rather than an incongruent one. Importantly, as context was matched between congruent and incongruent trials for this comparison (all correlations were restricted to those between trials that had been paired with the same background context), the observed increase in pattern similarity in the congruent over the incongruent condition must be due to congruency between the object and schematic context rather than due to overlapping perceptual similarities between the background contexts.

We next tested the hypothesis that pattern similarity would increase within context but not across context if representations were organized according to the paired schema. We computed a linear mixed model predicting pattern similarity as a function of context (same context/across context) and delay (short/long), separately for congruent and incongruent pairs. Random intercepts were included for each participant, with random slopes for counterbalancing condition, and weights were included to model unequal variance across fixed effects. We found that for congruent pairs (congruent within versus congruent across bars in Fig. 4b) there was a main effect of context, due to greater similarity of patterns within than across context ($F$(1,17746) = 6.68, $p$ = 0.010, M difference = −0.007, CI:[−0.013–(−0.002)]). There was also a main effect of delay, such that similarity became greater over time ($F$(1,17746) = 107.83, $p$ < 0.0001, M difference = −0.022, CI: [−0.027–(−0.017)]). There was no context by delay interaction ($F$(1,17746) = 1.52, $p$ = 0.218, M difference = 0.004, CI: [−0.003–0.012]). This indicates that patterns for objects that shared the same congruent context were more similar to each other than they were to congruent object-context pairs of the opposing context, irrespective of delay. For incongruent pairs (incongruent within versus incongruent across bars in Fig. 4b), although there was a main effect of delay ($F$(1,10380) = 12.37, $p$ = 0.0004, M difference = −0.010, CI: [−0.019–(−0.002)]) whereby similarity generally increased over time, there was no effect of context ($F$(1,10380) = 1.14, $p$ = 0.285, M difference = 0.002, CI: [−0.008–0.012]) and no context by delay interaction ($F$(1,10380) = 0.0007, $p$ = 0.979, M difference < 0.001, CI: [−0.11–0.11]). Thus, patterns for incongruent object-context pairs were becoming more similar over time but they were not being integrated according to the schematic context with which the objects were paired.

We ran several control analyses to complement the above findings. We found that integration of congruent object-context pairs over time is not driven by the fact that objects in the congruent condition are more semantically similar to each other than are objects in the incongruent condition (Supplementary Method 6; Supplementary Fig. 6). Thus, integration apparently proceeds according to overlapping congruent contexts rather than overlapping incongruent contexts or conceptual similarity between objects. Furthermore, lack of context-specific integration in the incongruent condition was not driven by inflated correlations in the across-context condition due to residual semantic overlap between objects and opposing contexts (Supplementary Method 7; Supplementary Fig. 7). We also examined integration in the mPFC restricted to detailed memory trials and similarly found evidence of increased representational overlap for schema-congruent information (Supplementary Method 8; Supplementary Fig. 8), suggesting that mPFC integration does not necessarily reflect loss of detail, and coarse features of detailed memory trials may also be integrated in the mPFC. In addition, integration of congruent

trials in the mPFC over time was not related to post-encoding connectivity between the anterior hippocampus and mPFC (Supplementary Method 9). We therefore, did not observe evidence that post-encoding interaction with the anterior hippocampus was related to subsequent organization of congruent memories in the mPFC. Further, the same set of pattern analyses applied to forgotten trials indicated that observed integration in the mPFC over time is specific to trials that were remembered (Supplementary Method 10; Supplementary Fig. 9a, Supplementary Table 2). Finally, we present pattern similarity plots for within-context correlations separately for beaches and kitchens in Supplementary Fig. 10 (Supplementary Method 11). Further research is required to investigate category-specific differences in representation.

## Neural specificity in the hippocampus for detailed episodic memories

As the hippocampus is required for the retrieval of specific episodic events, we next investigated whether the representation of detailed memories in the hippocampus varied as a function of mnemonic specificity and congruency. We ran a modified version of the pattern similarity analysis outlined above using only items for which both the context and the specific scene were retrieved. We focused on the anterior and posterior hippocampus as our ROIs, as it has been proposed that the posterior hippocampus represents fine-grained or detailed aspects of memory, while the anterior portion represents coarser aspects[5,40,42,53,54]. This time, we calculated three groups of correlations per successfully retrieved object-scene pair for each participant (Fig. 5a). For congruent pairs, we extracted the pattern of voxels within each ROI as participants were retrieving the scene associated with a given object. We then correlated the extracted pattern with 1) the patterns of all other objects in the congruent condition that had been paired with the same scene, 2) the pattern of all other objects in the congruent condition that had been paired with the similar scene of the same context, and 3) the pattern of all other objects in the congruent condition that had been paired with scenes from the other/opposing context. We did the same thing for incongruent pairs, except all correlations were between incongruent rather than congruent trials. For each ROI, we submitted these correlations to a scene (same/similar/other-context scene) × congruency (congruent/incongruent) × delay (short/long) linear mixed model, with a random intercept for each participant and random slopes for each counterbalanced group. We hypothesized that the posterior hippocampus would reflect scene specificity: patterns for objects paired with the same scene would be more similar to each other than to those for objects paired with the similar scene of the same context, as well as those for objects that had been paired with scenes from the opposite context. Conversely, we hypothesized that the anterior hippocampus would represent context but not scene specificity: correlations should be similar between representations for objects paired with the same and similar scene of the same context, but different from representations of objects that had been paired with scenes from the other context. We expected to see these patterns in the hippocampus at both delays irrespective of congruency if the hippocampus is important for representing detailed episodic memories regardless of content over time[55].

In the posterior hippocampus, we found a main effect of delay ($F$(1,16061) = 47.14, $p$ < 0.0001, M difference = −0.030, CI:[−0.042–(−0.017)]), driven by higher pattern similarity across the long delay than the short. There was also a main effect of scene ($F$(2,16148) = 5.00, $p$ = 0.007), which was driven by higher pattern similarity between objects sharing the same scene compared to those paired with a similar scene ($t$(16148) = 2.76, $p$ = 0.006, M difference = 0.010, CI: [0.003–0.018]) and compared to those paired with scenes from other contexts ($t$(16148) = 2.86, $p$ = 0.004, M difference = 0.009, CI: [0.003–0.016]). There was no difference in pattern similarity between the similar scene and other-context scene conditions ($t$(16148) = 0.30,

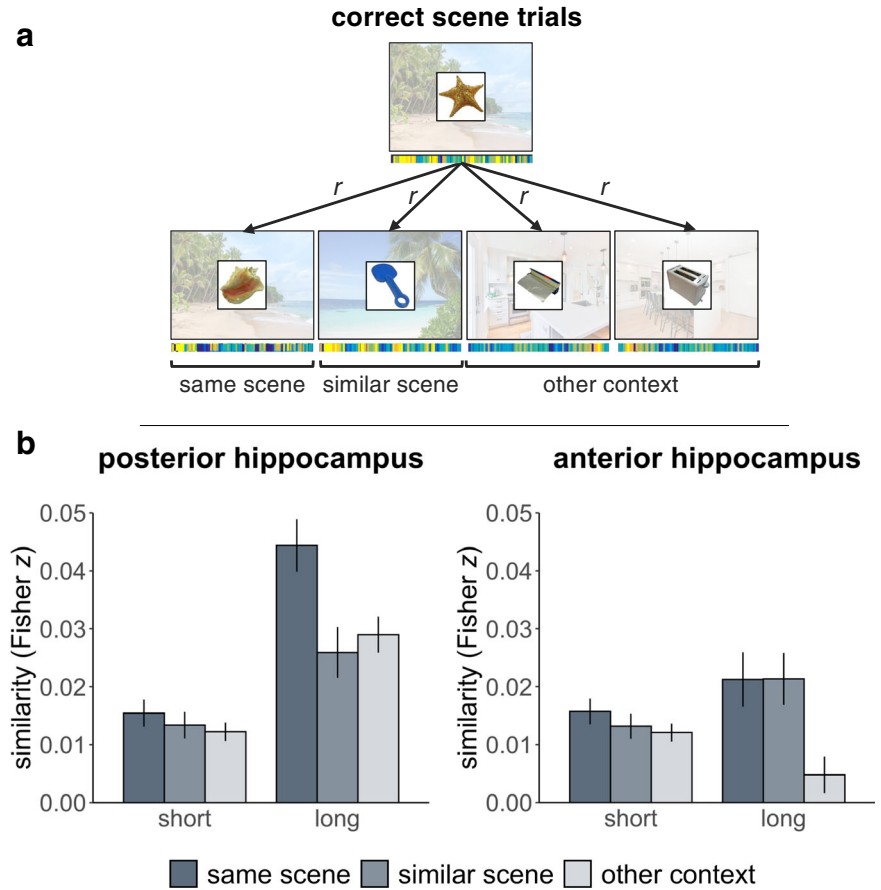

**Fig. 5 | Representational similarity analysis of object-scene pairs retrieved with detail in the hippocampus. a** Schematic example of our analysis approach. Patterns for successfully retrieved object-scene pairs were extracted from the right anterior and posterior hippocampus and correlated with objects that had shared the same scene, had been paired with the similar scene of the same context, as well as those that had been paired with scenes from the other context. This example depicts congruent correlations for one of the beaches, but the same analysis was applied to the other beach, as well as to both kitchens, within congruent and incongruent conditions. Background scenes were not presented during retrieval, but were retrieved from memory. **b** Resulting pattern similarity in the posterior and anterior hippocampus over time, according to scene/context overlap. Note that as congruency did not interact with scene condition, we plot the main effect of scene collapsed across congruency. Data reflect estimated marginal means from linear mixed effects models predicting pairwise Fisher transformed correlations from congruency, scene, and delay variables, in $N = 23$ participants across the short delay and $N = 19$ participants across the long delay. Errors bar reflect standard error of the mean adjusted for within-subject design. Scene and object images presented here are placeholders used for illustrative purposes. Objects were retrieved from the bank of standardized stimuli (BOSS) database (Copyright (C) 2009, 2010 Mathieu Brodeur)[100]. Beach photos by Rowan Heuvel and Pedro Monteiro, kitchen photos by Sidekix Media and Zac Gudakov, all on Unsplash: https://unsplash.com/license. Source data are provided as a Source Data file.

$p = 0.77$, M difference = −0.001, CI: [−0.007–0.005]). The rest of the effects in the model were not significant (Table 1), although the interaction between delay and scene was marginal ($F(2,16149) = 2.69$, $p = 0.068$), and a visual inspection of mean pattern similarity in each condition suggests the effect of scene was largely driven by pattern similarity across the long delay (Fig. 5b). These results indicate that when memories were retrieved with specificity, posterior hippocampal patterns distinguished between objects paired with the same scene from those that had been paired with similar and other-context scenes over time, irrespective of congruency. Thus, the posterior hippocampus differently represents fine-grained perceptual differences between scenes from the same schematic context. As there was no effect of congruency, we present the main effects collapsed across congruency in Fig. 5b for clarity, and the full un-collapsed data for hippocampal and mPFC ROIs in Supplementary Method 12/Supplementary Fig. 11. The same set of pattern analyses applied to forgotten trials did not yield any statistically significant effects (Supplementary Method 10; Supplementary Fig. 9b; Supplementary Tables 1 and 2). Further, when adding in coarse memory trials and collapsing across detailed and coarse memories (as done for the mPFC analysis), the effect of scene disappears, suggesting that representational scene

specificity is lesser in magnitude or absent for coarse memory trials (Supplementary Method 13; Supplementary Fig. 12). Finally, we did not observe the same pattern of representational scene specificity for detailed trials in the left posterior hippocampus (Supplementary Method 14; Supplementary Fig. 13a).

In the anterior hippocampus there was a main effect of scene ($F(2,16146) = 6.67$, $p = 0.001$), as well as a delay by scene interaction ($F(2,16147) = 3.78$, $p = 0.023$; Fig. 5b). The rest of the effects in the model were not significant (Table 1), although there was also a marginal interaction between congruency and delay ($F(1,16158) = 3.26$, $p = 0.071$, M difference = 0.014, CI[−0.006–0.033]; Supplementary Fig. 10). We unpacked the interaction between scene and delay with pairwise $t$-tests. At the short delay, there was no difference in pattern similarity between the three scene conditions (same scene vs similar scene: $t(16144) = 0.871$, $p = 0.38$, M difference = 0.003, CI: [−0.003–0.008]; same scene vs other-context scenes: $t(16143) = 1.44$, $p = 0.15$, M difference = 0.004, CI: [−0.001–0.009]; similar scene vs other-context scenes: $t(16144) = 0.45$, $p = 0.66$, M difference = 0.001, CI: [−0.004–0.006]). At the long delay, while there was no difference in pattern similarity between objects that shared the same scene versus similar scenes ($t(16147) = 0.02$, $p = 0.99$, M difference < 0.0001, CI:

**Table 1 | Results of hippocampal pattern similarity models**

| Effects | DFn | DFd | F | P |
|---|---|---|---|---|
| Posterior Hippocampus | | | | |
| Scene | 2 | 16148 | 5.00 | 0.007* |
| Congruency | 1 | 16153 | 1.06 | 0.30 |
| Delay | 1 | 16061 | 47.14 | <0.000001* |
| Scene × congruency | 2 | 16146 | 1.46 | 0.23 |
| Scene × delay | 2 | 16149 | 2.69 | 0.068~ |
| Congruency × delay | 1 | 16162 | 1.96 | 0.16 |
| Scene × congruency × delay | 2 | 16146 | 0.52 | 0.59 |
| Anterior Hippocampus | | | | |
| Scene | 2 | 16146 | 6.67 | 0.001* |
| Congruency | 1 | 16165 | 0.44 | 0.51 |
| Delay | 1 | 16062 | 0.60 | 0.44 |
| Scene × congruency | 2 | 16144 | 0.43 | 0.65 |
| Scene × delay | 2 | 16147 | 3.78 | 0.023* |
| Congruency × delay | 1 | 16158 | 3.26 | 0.071~ |
| Scene × congruency × delay | 2 | 16144 | 0.13 | 0.88 |

Scene (same/similar/other), congruency (congruent/incongruent), and delay (short/long) were included as factors in a linear mixed effects model predicting pattern similarity in the posterior and anterior hippocampi, with a random intercept for each participant and random slopes for each counterbalancing condition.

DFn degrees of freedom for the numerator, DFd degrees of freedom for the denominator.

*$p < 0.05$ and ~$p > 0.05 < 0.1$, uncorrected for multiple comparisons.

[−0.013–0.013]), pattern similarity was higher in both conditions compared to the other-context scenes condition (same scene vs other-context scenes: $t(16147) = 2.82$, $p = 0.005$, M difference = 0.017, CI: [0.005-0.028]; similar scene vs other-context scenes: $t(16147) = 2.85$, $p = 0.004$, M difference = 0.017, CI: [0.005–0.028]). Pattern similarity between objects that shared the same scene did not change over time ($t(16161) = 1.06$, $p = 0.29$, M difference = 0.006, CI: [−0.005-0.016]), nor did pattern similarity between objects sharing similar scenes ($t(16155) = 1.59$, $p = 0.11$, M difference = 0.008, CI: [−0.002-0.018]). There was a marginal decrease in correlations between patterns for objects from opposing scene contexts over time ($t(16155) = 1.95$, $p = 0.051$, M difference = −0.007, CI: [−0.015−0.00003]). These results indicate that over time the anterior hippocampus was not distinguishing between objects paired with specific scenes, but it was differentiating the coarser context regardless of congruency. The same set of pattern analyses run on forgotten trials did not yield any statistically significant effects (Supplementary Method 10; Supplementary Fig. 9c; Supplementary Tables 1 and 2). We did not observe the same pattern of representational context specificity in the left anterior hippocampus for detailed trials (Supplementary Method 14; Fig. 13b).

## Discussion

We examined the influence of real-world schemas on systems consolidation by probing memory quality, post-encoding hippocampal-mPFC functional interaction, and representation in the mPFC and hippocampus during subsequent retrieval. We found that only schema-congruent object-scene pairs were remembered more coarsely over 3 days, in line with evidence for a change in quality of memory due to increasing reliance on neocortical retrieval over time[2,3], as well as better retention of (and/or greater reliance on) schematic information when detail is forgotten[56]. We further showed that coarser quality of memory over time was associated with enhanced post-encoding coupling between the anterior hippocampus and mPFC. This finding is in line with theoretical work propounding the importance of offline functional interaction between these regions for updating established neocortical memory traces with consolidation[33,37,38]. Finally, we present evidence of greater representational overlap in the mPFC during the retrieval of schema-congruent than incongruent pairs with consolidation, despite the fact that context between these two conditions was matched. Furthermore, memory representations were specifically integrated within the paired congruent context, showing that schemas act as an organizing scaffold for the consolidation of congruent content. As we did not find evidence of neocortical integration across these three modalities of inquiry for incongruent pairs across the same time-frame, the totality of these findings accord with rodent work suggesting that schemas accelerate consolidation[12,19,23,24].

We investigated the nature of representations as they were influenced by delay, schematic congruency, and contextual overlap. Although pattern similarity in the mPFC increased over time during the retrieval of arbitrary associations that shared the same context – similar to what others have reported[41] – we demonstrated that schematic congruency not only enhances integration but also organizes neocortical representations according to existing knowledge structures. The fact that pattern similarity increased for both congruent and incongruent associations over time suggests that unique features may have been lost or minimized in both conditions over time. As associations were studied in the same broader experimental context (i.e., in the scanner, in the context of the same experiment), some degree of increase in pattern similarity that was not specific to the beach or kitchen stimuli may have occurred after a delay in both congruent and incongruent conditions[57]. It was only in the congruent condition, however, that memories became integrated according to the beach or kitchen context with which they were associated, either by strengthening overlapping elements[39,41] or by the distortion of common elements being pulled together in representational space[58–60], which is likely to occur with schematic assimilation[61]. Alternatively, congruent memories may have become more strongly linked such that retrieval of one pair reactivated other related pairs in the neocortex[34], thus increasing pattern similarity across trials. We postulate that the coarsening of memory observed for congruent object-context pairs over time is suggestive of one of the former interpretations. Finally, congruent memory traces may have been strengthened at encoding and therefore slower to decay, rather than actively schematized with consolidation. While a memory strength interpretation accords with the notion that strengthening overlapping neural ensembles enhances neocortical integration, the finding that schema-congruent mPFC representations are integrated only after a prolonged delay suggests extended consolidation processes are paramount. Moreover, we found evidence of schema-congruent neural integration in the mPFC even for detailed memories (Supplementary Method 8; Supplementary Fig. 8), which suggests that increased pattern similarity in the mPFC does not simply reflect degradation of detail. Likely, both encoding and post-encoding processes contribute to the lasting schema benefit to memory, as rodent work suggests[23,24].

Notably, while we observed that schematic congruency affected both quality of memory and neural integration in the mPFC, the fact that representational integration also occurs for detailed memories suggests that there is not a linear relationship between neural overlap in the mPFC and coarsening of memory. From the perspective of trace transformation theory, both neocortical and hippocampal memory traces can exist concurrently[2], thus coarsening of memory likely results from both consolidation of and reliance on neocortical elements, as well as loss or disuse of (likely posterior[9]) hippocampal contribution. In other words, an important additional contributor to memory transformation is likely lack of detail that depends on posterior representations retrieved by the hippocampus.

The fact that schematic context could be distinguished in the mPFC for only congruent pairs suggests that congruent object-context representations were organized according to the schema with which they were related. Thus, schematic beach and kitchen information was retrieved in the mPFC in response to congruent objects irrespective of delay. The mPFC was not representing schematic context in the

incongruent condition, despite the fact that contexts were ultimately retrieved. This lack of schema context effect is at odds with the finding by Tompary and Davachi[41] of increased representational overlap in the mPFC for arbitrary object-scene pairs within the same context relative to across contexts by one week, but it is possible that this effect emerges over longer time-frames with slower neocortical learning[17]. Further work is required, however, in both animals and humans, to demonstrate neocortical integration for both congruent and incongruent stimuli in the same study across different time-scales. An alternative explanation for the discrepancy between Tompary and Davachi's[41] findings and ours lies in differences in the encoding task instructions employed in each study. In their study, participants were asked to imagine the object interacting with the scene, which may have facilitated neural integration of the arbitrary object-scene pairings. In the present study, participants instead made congruency judgements during encoding that may have facilitated integration of congruent stimuli more than incongruent. We conjecture that integrative encoding and schematic congruency are two complimentary ways of achieving neural integration. The former, more effortful case, is likely necessary to build schemas and accommodate novel information while the latter likely facilitates integration of related content by leveraging existing information. It is plausible that incongruent stimuli in the present study would have shown stronger neural integration with an integrative encoding strategy.

We also used pattern similarity analysis to examine the granularity of representations for memories retrieved with specificity in the anterior and posterior hippocampus. Over time patterns in the posterior hippocampus came to differentiate objects that had been paired with the same scene from those paired with similar scenes and scenes from opposing contexts. Patterns in the anterior hippocampus did not differentiate between objects that had been paired with the same and similar scenes, but did differentiate between the broader kitchen and beach contexts. These findings follow rodent and recent human work indicating that by virtue of receptive field size, subfield composition, and functional and structural connectivity with the rest of the brain[42,48,49,51,62], the posterior hippocampus differentiates granular pieces of information in the service of episodic specificity, while the anterior hippocampus represents more global features such as episodic context[5,42]. Interestingly, this representational pattern only emerged after a delay.

It is plausible that visually similar overlapping information was representationally orthogonalized across the short delay, in line with the well-described role of the hippocampus in pattern separation[63]. Over time, overlapping information came to be integrated according to the degree of perceptual and contextual overlap, while also becoming differentiated from objects paired with other scenes and contexts. Several studies have reported representational changes in the hippocampus over time for overlapping or visually similar events[41,59,64–67]. Indeed, the most similar of these studies to ours found that pattern discriminability between overlapping and nonoverlapping object-scene pairs only emerged over time in the anterior and posterior hippocampus[41]. Furthermore, while a number of studies have been unable to decode mnemonic content in the hippocampus at relatively short delays[68–70], others have documented increasing accuracy in such decoding over time[71–73], in line with the present findings.

Computational modeling and rodent work indicates that the hippocampus can simultaneously represent orthogonal and overlapping information[74,75], but it remains to be specified under which circumstances the hippocampus integrates, orthogonalizes, or separates mnemonic content, as well as the scale of such processes in humans. Studies of representational similarity in the hippocampus have generally been mixed in this regard, and have rarely been investigated in terms of change over extended delays[58,76]. Our finding that hippocampal representations change with consolidation such that objects that share the same scene exhibit greater representational

overlap while simultaneously preserving scene and context information, agrees with the notion that consolidation serves to group common element together while also differentiating similar experiences. It follows that consolidation processes may be bidirectional; in addition to the hippocampus driving reorganization in neocortical networks, the opposite is likely true as well – possibly through neocortical-hippocampal-neocortical loops that act to consolidate memory during sleep or awake replay of learned content[77–80].

Finally, while there is theoretical work and empirical evidence that hippocampal engagement differs for congruent and incongruent information[16,81–83], how the hippocampus represents congruent versus incongruent content when it is engaged is much less clear (although see ref. 60). When detailed episodic memory was retrieved, we found evidence of representational specificity in the hippocampus regardless of congruency. These observations are in keeping with trace transformation theory, which posits that even though memory quality may become less precise over time with the establishment of – and reliance on – neocortical memory traces, detailed retrieval invariably involves the hippocampus[2,4], which operates as a high-fidelity relational binder of consciously apprehended information, regardless of content[55]. Notably, given the difficulty of obtaining enough coarse memory trials for reliable pattern similarity analyses, we have not shown that representational specificity is exclusive to detailed and not coarse memory trials. However, including coarse memory trials in our analysis (i.e., collapsing across coarse and detailed trials, see Supplementary Figs. 5 and 12) suggested that the effect of representational specificity is at least smaller for coarse trials. These analyses also indicated that congruency can influence pattern similarity in the hippocampus over time when coarse trials are included, which may indicate that congruency influences gist representation. Nonetheless, the finding that detailed memories are represented with specificity in the hippocampus regardless of congruency raises important considerations that have yet to be tested in the rodent literature: while schema-consistent information can be retrieved without the hippocampus relatively quickly, it is possible that such memory is nonetheless lacking the rich episodic detail and specificity the constitutes hippocampal memories[9]. Should task demands tax the memory system to retrieve such detail, we predict the hippocampus would be required – as demonstrated in rodent (and human) studies of memory in the absence of a learned schema[2,4,9,84].

To conclude, we present evidence that post-encoding hippocampal-mPFC coupling strengthens the coarse memory trace of schema-congruent memories after a prolonged period of consolidation. In parallel, we present evidence that real-world schemas act as organizing scaffolds that serve to enhance neocortical integration of related memories. The hippocampus, on the other hand, supported specificity of representation for detailed retrieval at the long delay irrespective of congruency. Interestingly, the pattern of hippocampal representation during retrieval evolved markedly over time and was suggestive of integration of overlapping content while simultaneously keeping similar memories distinct. This unexpected finding suggests that even detailed hippocampal representations change with consolidation, expanding the hypothesized role of the hippocampus to include the organization of contextual memory over time.

## Methods

### Participants

Twenty-three young adults (8 M/15 F, mean age: 26.39 years, range: 22–34) participated in this experiment. Sample size was determined by an a priori power analysis based on a study using a similar experimental design[41], and is outlined in Supplementary Method 15. Total percent correct was below chance (<33%) in four participants at the long delay, and, therefore, their data for the long delay was excluded in all analyses; data at the short delay were retained. For resting state connectivity analyses an additional 2 participants were excluded as

resting-state scans were not collected due to technical issues. Thus, the total sample size for the behavioral and pattern similarity analyses was 23 participants at the short delay and 19 at the long delay, while 17 participants were included in the resting-state connectivity analysis. All participants were English speakers with normal or corrected to normal vision, and no active diagnosis of neurological or psychiatric disorder. The experimental protocol was approved by the University of Toronto Research Ethics Board, and all participants provided informed consent.

## Experimental design

Participants underwent two fMRI sessions separated by ~72 h. During the course of the experiment participants underwent an encoding and cued retrieval session for object-scene pairs across a 10 min (short) delay, and again across a 72 h (long) delay. These delays were chosen to reflect long-term memory across a relatively short and extended delay. The 72 h delay was chosen based on behavioral piloting which indicated adequate performance across this timeframe, while also allowing multiple nights of sleep between study and test to provide the opportunity for extended consolidation processes to occur[79]. All encoding and testing took place within the fMRI scanner, and the order of the delays was counterbalanced to limit confounding practice effects and differences in neural similarity that could arise due to experience with the four scenes. Participants underwent fieldmap and structural scanning during the 10 min delay (i.e., they remained in the scanner), and went about their typical activities outside of the scanner during the 72 h delay.

The counterbalancing procedure resulted in two groups of participants with slightly different scanning procedures. In group A, the first scanning session involved encoding object-scene pairs, along with a cued-retrieval test for the learned pairs 10 min later. They would then encode a new set of object-scene pairs in the scanner, to be tested 72 h later during session 2. In group B, participants encoded object-scene pairs during the first session. During session 2, they were tested on the learned object-scene pairs (72 h delay), and then encoded a new set of stimuli, which they were tested on 10 min later. All participants were administered a learning and practice test (described below) prior to scanning to prepare for the main experiment within the scanner. After participant exclusions (detailed above), there were 10 participants in group A and 13 in group B for the behavioral and pattern similarity analyses. For the connectivity analyses there were 4 participants in group A and 13 in group B. We note here that random slopes for each counterbalancing group were fit in all behavioral and pattern similarity models to account for differences across the two groups. We present a plot of the connectivity results according to counterbalancing group in Supplementary Method 16/Supplementary Fig. 14, given the low sample size in Group A precluded accurate modeling of group differences for the connectivity analysis.

## Stimuli

Stimuli were presented using Inquisit 5 (Millisecond; https://www.millisecond.com/). Four scenic color photos (1920 ×1080 pixels) were used in this experiment: two beaches, and two kitchens. These beaches and kitchen scenes served as backgrounds to 160 pictures of objects in white boxes (300 ×300 pixels), 60 of which were objects typically found in kitchen contexts, 60 were typically found in beach contexts, and the remaining 40 were unrelated to either context (and were from a variety of other contexts). In order to minimize item-specific effects, objects were pseudo-randomly paired with each of the four scenes within and across congruent and incongruent contexts and delay for each participant, to construct two stimulus lists per participant: one for the short and one for the long delay. Each stimulus list consisted of 80 object-scene pairs, 40 of which were congruent (20 beach objects paired with beaches, 20 kitchen objects paired with kitchens) and 40 of which were incongruent (10 beach objects paired with kitchens, 10

arbitrary objects unrelated to either context paired with kitchens, 10 kitchen objects paired with beaches, 10 arbitrary objects unrelated to either context paired with beaches). Thus, congruency here refers to the relationship between each object and the schematic context with which it was paired. Half of the pairs in the incongruent condition consisted of objects typically found in the opposite context (e.g., oven mitts are typically found in kitchens, but were paired with a beach), in order to minimize the assumption that objects typically found in a context would always be paired with that context, and hence discourage the strategy of always choosing the congruent context during the memory test, described below.

## Training task

Given that we were interested in probing pattern similarity for scenes based on memory (and not based on re-exposure), it was important for the participants to learn the name of each of the four scenes thoroughly (Beach A, Beach B, Kitchen A, Kitchen B) so that they could later indicate which scene was paired with an object without being visually re-presented with the scene itself during memory testing. It was equally important to ensure that participants knew the difference between scenes of the same context so that they were not inadvertently indicating the wrong scene. Finally, we wanted to ensure that participants were able to visualize the scenes in detail during retrieval. To that end, participants underwent a training session 1 h prior to the first scanning session in which they learned the name of each of the four scenes, practiced visualizing the scenes in detail, and became acquainted with the task they were to undertake in the scanner. The training session consisted of 4 parts and took ~15–25 min to complete.

Part 1 of the training session was a self-paced format in which participants viewed each scene with its corresponding name, one at a time, and then all four scenes on the screen at once so that they could compare them. They were asked to pay attention to the name and to the visual details of each scene, so that they would be able to name and visualize them in detail later on. In part 2 participants were shown each scene one at a time and were asked to choose the name of the scene from the four available options. They were given immediate feedback as to whether they were correct or incorrect. If they were incorrect, they repeated the process until all of the scenes were correctly named. In part 3 they were given the name of each scene one at a time, and were asked to visualize the scene in as much detail as possible, as well as to rate how vivid their visualization was on a four point scale ranging from "could not visualize" to "vivid visualization". Right after visualization of a given scene they were asked one question pertaining to a detail of the scene (e.g., is the dishwasher located to the right or the left of the stove?), and had to select the appropriate answer out of two options. If the participant indicated they had a less than "good" visualization for any of the four scenes, or if they got any of the detail questions wrong, they re-studied the images and tried again with new questions about the details of the scenes (again, one question per scene). They repeated this process until they could produce good visualization and correctly answer the detail question for all scenes. In part 4 they underwent a practice encoding and cued-retrieval procedure as they were to be undertaken for the actual experiment for a small subset of images (12 object-scene pairs), as described in further detail below. Participants were given an abridged version of the training task before entering the scanner during the second session, wherein they completed Part1 and Part2 once more. This procedure served to ensure that the participants correctly remembered the name of each scene, and thus could proceed with the fMRI task.

## Encoding

For each encoding session, participants were presented with images of 80 objects one at a time, each paired with one of four background scenes, with pairings as described above (i.e., half congruent and half incongruent). The verbal labels associated with each background

scene were not present during encoding. Participants' task during encoding was to indicate if each object was related to the background scene or not. An object was to be considered related to the scene if the participant thought they might find the object in that context in real life. Participants were aware that they would be tested for their memory of the object-scene associations.

For each delay, they studied each object-scene pair three times across three 6.5 min long encoding runs. All pairs were presented in each encoding run in a pseudo-random order for each participant, such that adjacent trials did not share the same scene. Participants viewed each scene for 0.1 s on its own before it was overlaid with the paired object for an additional 2 s. This brief temporal overlaying strategy was implemented to emphasize that the object and scene were separate entities rather than a unitized construct. They were then presented with a screen with response options for 1 s during which they indicated if the object-scene pair had been related or unrelated using an MRI compatible button box. The response window was followed by a jittered fixation period lasting 1, 1.5, or 2 s.

### Cued retrieval
After each delay (10 min, 72 h), participants underwent a cued retrieval session during which they viewed studied objects individually in the absence of the paired background scene, and were asked to retrieve the scene that had been paired with the object as vividly as possible. The 80 learned pairs were tested across four 4-min runs with 20 objects presented in each run. Each object was presented for 2 s during which time participants were to visualize the paired scene. Participants were then shown a response screen and had 2 s to indicate with which context the object had been paired with (kitchen/beach/don't know). The response screen remained on for the duration of the 2 s regardless of the speed of the button press. If they indicated that the object had been paired with a kitchen or a beach, they were then shown another response screen for an additional full 2 s, during which they indicated with which specific beach or kitchen scene the object had been paired with (for example, if they chose "beach" they were offered the following response options: Beach A/Beach B/don't know). Piloting had revealed that some participants tended to over-rely on the "don't know" option, so they were instructed to use this option only when they had no memory of the correct answer, in lieu of guessing (i.e., they didn't have to have high confidence, but they should not guess). Objects were presented in a random order for each participant, and all responses were recorded using an MRI-compatible button box. Each trial ended with a jittered fixation period lasting 3–6 s.

### Behavioral data analysis
All statistical testing was performed using RStudio version 1.2.5033 (RStudio Team, 2019; http://www.rstudio.com/)[85]. To examine the influence of prior knowledge on memory over time, congruency of the object-scene pairs (congruent/incongruent) was scored based on each participant's judgments during encoding. Given that participants viewed each object-scene pair three times during encoding, this decision was operationalized as concordance on at least two of the encoding trials. On average, participants judged approximately half of the object-scene pairs at encoding as congruent and half as incongruent at both the short (congruent trials: M = 40.30, SD = 1.52; incongruent trials: M = 39.70, SD = 1.52) and long (congruent trials: M = 40.84, SD = 1.57; incongruent trials: M = 39.05, SD = 1.61) delays. Participants were generally consistent in their judgements of trials as related or unrelated across encoding runs (M = 81.58% of trials were consistent, SD = 13.50%), indicating that they were not responding randomly. Total percent correct retrieval based on congruency was calculated as the percent of total congruent or incongruent trials where participants identified the correct context (regardless of the specific scene) separately for congruent and incongruent pairs at each

delay (short/long). In the examination of memory granularity, we defined detailed memories as the percent of objects for which participants correctly retrieved both the context (beach/kitchen) and the specific scene (e.g., Beach A/Beach B). We defined coarse memories as the percent of congruent and incongruent encoding judgements in which participants correctly identified the context with which an object had been paired with (beach/kitchen), but indicated that they did not know the specific scene, or else chose the incorrect but visually similar scene during retrieval. Differences in memory retrieval between conditions were tested using linear mixed effects models with a random intercept for each participant and random slopes for each counterbalancing condition, using the lme4 package[86] (version 1.1.30; https://cran.r-project.org/web/packages/lme4/index.html). Denominator degrees of freedom and *p*-values were estimated using the Satterthwaite approximation as implemented using the lmerTest package in R[87] (version 3.1.3; https://cran.r-project.org/web/packages/lmerTest/index.html), given that this method produces results with relatively low Type I error rates and gives the most comparable results to regular linear models[88]. Normality of model residuals was assessed by inspecting histograms of the residuals and with Shapiro Wilk's tests, as implemented in the base stats package in R (version 4.2.1). In keeping with the assumptions of linear mixed models, if model residuals did not follow a normal distribution, the dependent variable was transformed by taking the square root, and the model was re-run on the normalized data. Homogeneity of variance of the model residuals was assessed for each model by visually inspecting a plot of the model residuals versus fitted values, and using Levene's test for unequal variance, as implemented in the car package in R[89] (version 3.1.0; https://cran.r-project.org/web/packages/car/index.html). If Levene's test indicated heteroskedasticity, we re-ran the model with a specified variance structure using the nlme package (version 3.1.158; https://CRAN.R-project.org/package=nlme), to allow the variance to vary across levels of the heteroskedastic fixed effects[90]. Confidence intervals reflect 95% confidence as determined via bootstrapping using the confint function of the stats package in R (version 4.2.1). We reported, plotted, and tested the raw descriptive means for each condition.

### fMRI parameters
All scanning was performed using a Siemens Prisma 3T full-body MRI scanner. Visual stimuli were projected onto a screen that was viewed through a mirror attached to the head coil. Functional echo-planar imaging (EPI) scans were oriented horizontally to intersect the anterior and posterior commissures (TR = 1.5 s TR, TE = 26 ms, flip angle = 70°, FOV = 220 × 220, 52 slices, 2.5 mm × 2.5 mm × 3 mm voxels), and were acquired with a GRAPPA acceleration factor of 1, and a multiband factor of 2. Phase encoding was in the anterior to posterior direction, with interleaved acquisition in the inferior to superior direction along the z-axis. A fieldmap scan was also collected, using a double-echo gradient echo sequence with the same parameters as the EPI sequence (with the exception of the following: TR = 0.88, TE1 = 4.92 ms, TE2 = 7.38 ms, flip angle = 60°). A T1-weighted magnetization-prepared rapid-acquisition gradient echo (MPRAGE) sequence (1 mm isotropic voxels, 160 sagittal slices) was also collected.

### Regions of interest definition
The hippocampi were anatomically defined for each participant using FSL's automatic subcortical segmentation protocol (FIRST). We chose to focus on the right hippocampus given its sensitivity to visual memory[91], but we present pattern similarity data from the left hippocampus in Supplementary Fig. 13 for the interested reader. Each participant's hippocampi were manually segmented in native space along the long axis at the uncal notch to create anterior and posterior hippocampal ROIs[42]. The mPFC mask was constructed from combining areas A14m and A10m from the Brainnetome atlas[92] bilaterally in MNI space (https://atlas.brainnetome.org/). These ROIs are together

relatively inclusive of the mPFC. We did not include some of the most ventral mPFC ROIs of the Brainnetome atlas due to a high degree of signal dropout in these areas in some of our participants, resulting in noisy signal. The resulting mPFC mask was warped into each participant's native space using FSL's FLIRT function.

## Resting state connectivity analysis

Pre- and post-encoding resting state scans were acquired during session 1. The baseline resting state scan was acquired at the beginning of the scan. Given that our hypotheses pertained to quality of memory over time, and given that Tompary and Davachi[41] found that anterior hippocampus - mPFC connectivity was associated with representation of remote memories, we were specifically interested in changes in connectivity from baseline to post-encoding for stimuli that were to be tested after the long delay. The placement of the post-encoding resting-state scan occurred, therefore, directly after all three encoding runs for stimuli to be tested across the 72 h delay (there was no resting state scan after encoding stimuli to be tested across the short delay). Rest scans were 6 min long, wherein participants were instructed to fixate on a small black cross in the center of a gray screen and remain awake.

Resting state scans were used to measure encoding-related changes in functional connectivity between the mPFC and the anterior hippocampus, as indexed by correlations between low frequency fluctuations in BOLD activity of each ROI[45]. We chose the right anterior hippocampus because this region has greater structural and functional connectivity to the mPFC than the posterior hippocampus[42,48–50], and because connectivity between the anterior hippocampus and mPFC has proven to be related to mnemonic representation and retrieval of remote memories[41,51]. The resting state scans were preprocessed and modeled as separate sessions using CONN version 18b[93] (https://web.conn-toolbox.org/), which utilized the Statistical Parametric Mapping 12 (SPM12; https://www.fil.ion.ucl.ac.uk/spm/software/spm12/) toolbox via MATLAB R2016b (Mathworks) for preprocessing. The first 6 volumes were removed to allow for scanner stabilization. Motion was estimated and realignment, unwarping, and distortion correction were applied to the EPI images simultaneously. Volumes contaminated by sudden large head movements were identified using the Artifact Detection Toolbox[93] (ART), which flagged TRs with fluctuations in global signal greater than 3 standard deviations, translational motion greater than 1 mm, and rotational motion greater than 0.05 radians. The EPI images were co-registered to the T1-weighted anatomical scan, and were segmented into gray matter, white matter, and cerebrospinal fluid masks for each participant. We used aCompCor[94] to exclude physiological noise by regressing out the top five principal components from the data – as identified from a principal components analysis on the unsmoothed signal from eroded white matter and cerebral spinal fluid masks. The motion parameters (6 rigid body realignment parameters and their first order temporal derivatives, plus the high motion volumes identified by ART) were also regressed out, and the data were temporally filtered to exclude very low (<0.008 Hz) and high (>0.09 Hz) frequency fluctuations.

Average timeseries across the unsmoothed voxels within each native-space ROI were used to compute a Pearson's correlation between the ROIs of interest for each participant (mPFC- anterior hippocampus). Correlation values were Fisher transformed, and the resulting values from the pre-encoding scan were subtracted from the post-encoding values for each participant. These post-pre difference scores in pairwise connectivity for each participant were then correlated with participants' percent correct retrieval scores (congruent/incongruent coarse/detailed retrieval), using a one-tailed test for our a-priori hypothesis, and two-tailed tests for exploratory correlations, using the base stats package in R (version 4.2.1). Confidence intervals reflect 95% confidence. Tests for differences between correlations (Supplementary Method 4) were conducted using William's test for dependent correlations using the psych package in R (version 2.2.5; https://cran.r-project.org/web/packages/psych/index.html).

## Pattern similarity estimation

All retrieval scans were preprocessed using FSL (FEAT; http://www.fmrib.ox.ac.uk/fsl). Encoding scans were not analyzed for the present manuscript. The first 6 volumes of the EPI images were removed to allow for scanner stabilization. For each functional run, head movement was estimated (6 rigid body motion estimates corresponding to translations and rotations around x, y, and z-axes, which were saved as regressors for later modeling) and the EPI was realigned to correct for motion. Volumes with framewise displacement >0.9 were flagged, to be used as regressors during first level modeling in order to account for large changes in signal intensity that occur with sudden large head movements[95]. To reduce spatial distortion of the EPI images, an unwrapped phase map in rad/s was constructed from the magnitude (skull-stripped) and phase fieldmap images, and applied to the EPI data simultaneously with motion correction to minimize interpolation-related image blurring. Co-registration of the EPI image to the skull-stripped T1-weighted anatomical image was also performed during this step using boundary-based registration (BBR). The EPI images were smoothed with a 3 mm FWHM Gaussian kernel. All analyses took place in native space.

All preprocessed retrieval scans were modeled in each participant's native space. We took a Least Squares Single (LSS) pattern estimation approach[96,97], wherein each trial's activation was estimated with a separate GLM. The first regressor in each model represented the trial of interest (specifically, the portion of the trial where the object was on the screen and the participant was remembering the paired associate), and five additional regressors modeled the remaining trials within the same run according to trial type (coarse congruent, coarse incongruent, detailed congruent, detailed incongruent, forgotten). There were additional regressors for each response window. Finally, in order to correct for head motion, there were 6 regressors for rigid body motion parameters (translations and rotations around x, y, and z-axes), as well as a regressor for each TR that was flagged as having greater framewise displacement than 0.9 during preprocessing[95]. Regressors were convolved with a double gamma HRF. A map of t-values for the first parameter estimate was retained for each model and represents the activation for each trial during retrieval. For each trial, the spatial pattern of activity across each ROI was extracted into a vector and z-scored. Similarity between different vectors was calculated with Pearson correlations, which were Fisher-transformed prior to statistical testing. To avoid inflated correlations due to temporal proximity within each run, correlations were limited to trials occurring in different runs[97].

## Pattern similarity correlations for congruency dimension

At each delay (short/long) correlations were computed on mPFC patterns between objects that shared the same context (within kitchen or beach; within-context correlations), as well as between patterns for objects that had been paired with opposing contexts (kitchen vs beach; across-context correlations), separately for congruent and incongruent pairs. Specifically, similarity was computed for every retrieval trial in which the context was correctly retrieved (beach/kitchen) regardless of whether the specific scene was correctly identified (Beach A/Beach B or Kitchen A/Kitchen B). We included both coarse and detailed trials in order to increase statistical power, because presumably, coarse information should be retrieved for both trial types (e.g., general features of beaches). We note here that this analysis therefore should capture the representation of coarse features of memory (regardless of memory quality), which does not necessarily correspond to the phenomenological experience of coarse quality of memory. In the congruent condition, the retrieval vector of each

congruent trial (i.e., trials where the object had been congruent with its paired context) was correlated with the retrieval vectors of all other objects that shared the same context (and were also congruent with that context), as well as with the retrieval vectors of all other objects that were paired with the opposing context (and were congruent with that context). Similarly, in the incongruent condition, the retrieval vector of each incongruent trial (trials where the object had been incongruent with its paired context) was correlated with the retrieval vectors of all other objects that shared the same context (and were also incongruent with that context), as well as with the retrieval vectors of all objects that had been paired with the opposing context (and were also incongruent with that context). We additionally confirmed that differences in univariate activation between congruent and incongruent pairs over time were not driving pattern similarity results in the mPFC, as described in Supplementary Method 17.

### Pattern similarity correlations for scene granularity dimension
At each delay and for each ROI (anterior hippocampus/posterior hippocampus), we computed a series of correlations between trials that had been remembered in detail, depending on congruency and the specific scene the object had been paired with. Retrieval similarity was computed for every retrieval trial in which both the context and the specific scene that had been paired with the object was correctly retrieved (i.e., detailed memories). For congruent object-scene pairs, each trial's retrieval vector was correlated with 1) the retrieval vector of all other objects in the congruent condition that had been paired with the same scene, (same scene correlations), 2) the retrieval vector of all other objects in the congruent condition that had been paired with the visually similar scene of the same context (similar scene correlations), and 3) the retrieval vector of all other objects in the congruent condition that had been paired with scenes from the opposite context (other-context scene correlations). For the incongruent trials, we ran the same correlations except all of the correlations were between incongruent object-scene pairs (again, depending on whether the objects were paired with the same scene, visually similar scene, or other-context scenes). We additionally confirmed that differences in univariate activation between detailed congruent and incongruent pairs over time were not driving pattern similarity results in the hippocampus, as described in Supplementary Method 17.

### Statistical testing of pattern similarity
Statistical testing was performed using RStudio version 1.2.5033 (RStudio Team 2019; http://www.rstudio.com/). All correlations were Fisher transformed before being submitted to statistical tests. Individual pairwise correlations were plotted and inspected across conditions for each ROI to assess for outliers that could be driving significant effects (Supplementary Method 18, Supplementary Fig. 15). Trial-level similarity was estimated using linear mixed effects models with a random intercept for each participant and random slopes for each counterbalancing condition, using the lme4 package[86] (version 1.1.30; https://cran.r-project.org/web/packages/lme4/index.html). In keeping with the assumptions of linear mixed models, normality of each model's residuals was confirmed by inspecting a histogram of the model residuals and with Shapiro-Wilks tests of normality, as implemented in the base stats package in R (version 4.2.1). Homogeneity of variance of the model residuals was assessed for each omnibus model by visually inspecting a plot of the model residuals versus fitted values, and using Levene's test for unequal variance, as implemented in the car package in R[89] (version 3.1.0; https://cran.r-project.org/web/packages/car/index.html). If Levene's test indicated heteroskedasticity, we re-ran the model with a specified variance structure using the varIdent function and the weight argument using the nlme package (version 3.1.158; https://CRAN.R-project.org/package=nlme), to allow the variance to vary across levels of the heteroskedastic fixed effects[90]. As models containing unequal variance across three-way interactions

were too complex for the models to resolve with the added variance structure, these models were broken down into simpler models, testing 2-way interactions of interest such that the unequal variance could be appropriately modeled. Denominator degrees of freedom and p-values were estimated using the Satterthwaite approximation as implemented using the lmerTest package in R[87] (version 3.1.3, https://cran.r-project.org/web/packages/lmerTest/index.html), given that this method produces results with relatively low Type I error rates and gives the most comparable results to regular linear models[88]. We note here that the denominator degrees of freedom for hierarchical models are based on the number of level 1 observations, which in our pattern similarity models corresponds to the number of pairwise trial-level correlations being modeled across all conditions. Confidence intervals reflect 95% confidence as determined via bootstrapping using the confint function of the base stats package in R (version 4.2.1). Due to differing amount of data in each condition, we reported and plotted the estimated marginal means (also known as adjusted means, extracted using the emmeans package in R, version 1.7.5: https://cran.r-project.org/web/packages/emmeans/index.html). Estimated marginal means are calculated by giving equal weight to each cell in the model, and are, therefore, unbiased by imbalances in the data; in other words, they estimate what the marginal means would be had there been equal trial numbers in each condition. Main effects and interactions were interrogated using pairwise t-tests on the estimated means from each omnibus model, using the emmeans package (version 1.7.5). Within-subject error bars were computed for plotting purposes using the Morey (2008) method[98] using the Rmisc package in R (version 1.5.1; https://cran.r-project.org/web/packages/Rmisc/index.html).

### Reporting summary
Further information on research design is available in the Nature Research Reporting Summary linked to this article.

## Data availability
The raw neuroimaging data are protected and are not available due to data privacy laws. The processed behavioral data, connectivity data, and pattern similarity data are freely available on Zenodo (https://doi.org/10.5281/zenodo.6980915)[99]. Source data for all figures are provided with this paper. Source data are provided with this paper.

## Code availability
Code to plot Figs. 2–5 and to reproduce all statistical models in the manuscript has been deposited on Zenodo and can be found at: (https://doi.org/10.5281/zenodo.6980915)[99].

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

## Acknowledgements

We would like to thank Leanne Fernandes for her assistance with data collection, Dr. Katherine Duncan and Anuya Patil for sharing their fMRI sequence with us, Drs. Margaret Schlichting, Morris Moscovitch, and Alexander Barnett for their comments and advice during various stages of this project. This work was supported by the Natural Sciences and Engineering Research Council of Canada Research Grant #RGPIN-2015-06471 (M.P.M.) and a stimulus grant from the Toronto NeuroImaging (ToNI) facility at the University of Toronto, Department of Psychology (M.P.M. and S.A.).

## Author contributions

Conceptualization, S.A. and M.P.M. Methodology, S.A. and M.P.M.; Software, S.A.; Formal Analysis, S.A.; Investigation, S.A.; Resources, M.P.M.; Writing – Original Draft, S.A.; Writing – Review & Editing, S.A. and M.P.M.; Visualization, S.A. and M.P.M; Supervision, M.P.M.; Funding Acquisition, S.A. and M.P.M.

## Competing interests

The authors declare no competing interests.
