## [Peer Review File · Nature Communications]

Schemas provide a scaffold for neocortical integration of new memories over timeREVIEWER COMMENTS

Reviewer #1 (Remarks to the Author):

Summary:

In this manuscript, Audrain and colleagues examine how semantic congruency and contextual overlap influence associative memory across time, while also exploring how such memories are represented in the hippocampus and mPFC. Participants were scanned while they encoded object-context pairs that were either congruent or incongruent with semantic knowledge. They then retrieved these associations either after a short (10 minute) or long (3 day) delay. The granularity of retrieved memories was measured by testing whether participants remembered only the general context paired with each object (beach vs. kitchen) or the specific scene image that was studied (e.g., big beach vs. small beach). Behaviorally, the authors found that when object-context associations were semantically congruent (e.g., beach—seahorse), participants were more likely to remember them at a coarse level of granularity after a long vs. short delay. This same effect was not observed for incongruent associations, taken to reflect that only memories consistent with existing knowledge schemas had grown increasingly abstract over time. Several neuroimaging results were highlighted: (1) First, increased coupling between anterior hippocampus and mPFC after encoding was associated with a greater percentage of coarsely-remembered congruent associations after a delay. (2) In mPFC, retrieved memory representations were more similar between pairs that shared the same context (vs. different contexts) — but only when object-context associations were congruent with semantic knowledge. (3) After a delay, memory representations in the posterior hippocampus were more similar when they shared the same specific scene, whereas the anterior hippocampus clustered representations according to their shared context, but did not differentiate between specific scenes. These imaging results collectively demonstrate that prior knowledge and contextual overlap influence how memories come to be represented in the brain: whereas the mPFC groups memories primarily based on their accordance with an existing schema, the hippocampus organizes memories based on contextual/representational overlap at multiple levels of granularity.

The results extend previous work to examine how overlapping long term memory associations are influenced by the congruency of encoded information. Examination of how congruent compared to incongruent associations are encoded and perhaps differentially consolidated is of critical importance in our understanding of memory transformations. These results will add to the growing literature in this area. There are, however, clarifications of the bases for some conclusions made in the paper.

Major concerns:

- The authors frequently reference how their results provide support for “accelerated” consolidation. Although these findings clearly show that schematic knowledge and/or contextual overlap influence the transformation/organization of memories over time, it’s less evident how they establish acceleration *per se*. The most compelling result in support of this idea seems to be that the percentage of coarse incongruent memories was marginally greater during the long vs. the short delay, but not to the same degree as congruent memories (suggesting that these incongruent pairs might eventually be abstracted in the same way that congruent pairs were). But it’s still unclear how the mPFC similarity results speak directly to this idea, as the discussion suggests. (Unless perhaps we compare the present results with those of Tompary & Davachi, 2017, which do show evidence of integration between incongruent associations that share the same context, but only after a week-long delay. While this paper is mentioned in the manuscript, it isn’t described as critical for the idea of acceleration?) The authors should either explain the rationale more clearly behind this conclusion, or revise their claims about acceleration in the Intro/Discussion.

- The rationale behind why the correlation between ant. hipp—mPFC connectivity and coarse memory performance *must* reflect a consolidation mechanism (rather than something like memory specificity) is somewhat confusing. It’s argued that this conclusion stems from the fact that there was no such correlation between post-encoding connectivity and coarse memory at the short delay. However, the post-encoding rest periods in this study followed the encoding of pairs that were tested after the long delay — not the short delay. Particularly for participants in Group B (where rest scans were collected on an entirely different day than when the short delay pairs were studied), this conclusion feels unwarranted, as rest periods could not feasibly have captured consolidation-related activity for these short delay pairs.

- In the mPFC similarity analyses, it sounds like the pairwise trial comparisons that go into the computation of the “incongruent across-context” bin will include some trials in which there is a semantic relationship between the scene in one pair and the object in another (e.g., “beach—spoon” vs. “kitchen—seahorse”). Although this would only be true of a subset of these pairs, could it contribute at all to the lack of a difference between within vs. across-context associations in the incongruent bin? (perhaps because some of the across-context comparisons actually include pairs with semantic overlap, thereby boosting similarity?)

- All pattern analyses seem to have been conducted on trials with accurate performance (whether coarse or fine-grained). Assuming there are enough trials to examine (at least within the long delay timepoint), has it been verified that these effects do not emerge when memory is unsuccessful?

Minor concerns:

- Because the data in Figure 4b are meant to summarize results obtained from 3 separate models, it would be helpful if the authors could direct attention to which particular bars on the graph reflect the important comparisons in each model. For example, the first model described in the text tells us that the two “congruent within” bars (long vs. short delay) show a greater difference than the two “incongruent within” bars. As-is, the text is somewhat challenging to connect to the figure.
- Why might similarity in mPFC and the hippocampus between pairs in all conditions be going up over time, even when there is no content overlap? The authors could offer some speculative explanations in the discussion: for example, maybe the fact that all associations were studied in the same general context (e.g., an fMRI experiment) led them to become more similar after a delay (similar to what has been previously linked to anterior hipp-mPFC connectivity in Cowan et al., 2021).
- The distinction between contextual similarity vs. semantic/schematic similarity feels at times less clear-cut than the authors suggest. Although the relationship between a context (beach) and an item (seahorse) is certainly more abstract/high-level than the relationship between two different beaches, both rely on our ability to generalize across various exemplars linked to the same underlying semantic construct (beaches). Is the key distinction in this study that only contextual overlap involves any notable perceptual similarity between pairs? Clarifying the ideas here might help shed light on why the hippocampus is apparently agnostic to semantic congruency in the present study.

Reviewer #2 (Remarks to the Author):

Please see attached file.

The manuscript addresses an important and timely question, namely how prior knowledge influences memory transformation over time. This is an interesting paper, linking together exciting psychological and neuroscientific questions and using an elegant and suitable task and fMRI methods to answer these questions. Further, conducting such a complicated multi-session fMRI study is a herculean effort which should be recognized. The Introduction is beautifully written - clear and concise, and the fMRI methods are largely solid. However, I do have concerns about some aspects of the design and the logic of some of the analyses. Further, the specificity of the results should be clarified to evaluate their meaning and significance. I detail these concerns below.

Major concerns:

1. I am confused about some choices made for the RSA analyses, and how to interpret them considering the behavioral effects. The main behavioral result emphasized by the authors is the increase in coarse memory over time, specifically for congruent pairs. The authors interpret that as reflecting schematization of congruent items with time and the loss of details. The resting state connectivity analyses indeed speaks to that result. However, the RSA analysis addressing congruency collapses across detailed and coarse memories. And, collapsed across both levels of memory, the authors interpret this result as integration/schematization as well. The analyses in the hippocampus are only for detailed memories. It is unclear to me what motivates these choices for the RSA analyses, or how the RSA analyses inform us about the behavioral effect. If integration of the neural representations of congruent items lead to forgetting of details, should we observe higher mPFC similarity only for coarse memories, but not detailed memory? Or do the authors predict that neural integration should be only based on congruency, regardless of the details in memory (and if so why, given the behavior)?
2. Somewhat relatedly, while the authors refer to their results in terms of memory or mnemonic representations, the analyses do not address memory, in the sense that there is no comparison between different levels of memory (i.e., remembered associations vs. forgotten, or detailed vs. coarse memories). Thus, it is unclear to what extent the results reported in the paper reflect participants' memory. I am not sure whether the authors have enough items to run a meaningful analysis (see below as well), but it might be that at least for congruent-long delay, which is the main interest of the mPFC analysis and a focus of the study, there might be enough to run a meaningful analysis. Likewise, for the hippocampus, where there is no difference based on congruency, the authors might be able to collapse across congruent and incongruent items, to have enough items to compare detailed and coarse memories. Especially in the posterior hippocampus, where the argument is about detailed memories, we should expect the effect in the delayed condition to be specific to items remembered in detail, but not to coarsely remembered items. If the result reported in the posterior hippocampus appears in both detailed and coarse memories, I believe the interpretation should be different, as this similarity does not correlate with detailed memories. The authors should provide such comparisons between memory levels, to the best of their ability within the constraints of the design. If this cannot happen, please refrain from relating to the findings as necessarily reflecting memory, as we do not know how they differ by memory status. Importantly, even if unrelated to memory, at least in the mPFC the

results still reflect transformation of congruent vs. incongruent information over time (and in the hippocampus, transformation of information learned within same context over time, regardless of congruency). This is interesting and novel by itself, although we just might not learn from this study how these transformations relate to participants' memory for this information.

3. If I understand correctly, the authors interpret the increase in neural similarity as integration or schematization of congruent items (which is potentially reflected in forgetting of details, see also comment 1). Another interpretation could be that similarity merely reflects forgetting of details. Since there is an increase in coarse memory across the delay in congruent items, that could potentially lead to increased similarity only in congruent items. It seems to me like comparing specific levels of memory could alleviate this concern as well: if similarity only reflects loss of details, congruent and incongruent items remembered with the same level of specificity (coarse, or detailed) should not show a difference in similarity. If, however, similarity is driven by congruency, congruent items should be more similar even when comparing similar levels of memory for details.
4. One potential confound in the current study, at least for connectivity and the mPFC results, is participants' responses. Congruent items were responded with 'related', whereas incongruent with 'unrelated'. I understand this choice from the perspective of the congruency paradigm because many previous studies have used the same task (though note that some previous studies have equated responses). However, previous studies that have used related/unrelated responses, typically compared between different levels of memory (within the same response), which alleviate this concern. Here, it is possible that integration in mPFC, as well as functional connectivity with the hippocampus, is a result of integrative encoding promoted by considering items as 'related', rather than by congruency. This possibility is consistent with the lack of a replication of Tomparry et al., 2017. In their study, integrative encoding was required by the task, and they observed integration even for items that are incongruent (across a longer delay, as the authors note). Naturally, we cannot know the source of the discrepancy based on these two studies alone. To be clear, I am also not concerned by the lack of replication itself. To answer this concern as well, showing a difference between detailed and coarse congruent memories would be most convincing. Otherwise, the authors should consider in the Discussion that their results might be driven by the nature of the task, rather than congruency.
5. All the analyses collapse across both kitchen and beach scenes. The authors should include that as a factor in their analyses (for example, by including category in their statistical models) and see that the results hold when considering the different scenes, and do not stem from one scene only. It would also be useful to plot the results by category of scenes in the supplementary to. On that note, the control analysis presented in Supplementary Figure 6 is very important and a wonderful addition to the paper. I was also worried that since most of the items in the experiment were from either kitchen or beach, it could be that time-dependent integration in mPFC is promoted for the most prevalent or salient categories. This, however, is clearly not the case. It is interesting that the arbitrary objects also lead to integration (like congruent objects). This replicates Tomparry et al. (2017), but also suggests that some interaction between semantic category and incongruency

influences consolidation and integration, which is very interesting. The authors might want to address this in the Discussion.

6. For all the results reported in the manuscript – the authors should consider group as a factor in the analysis (e.g., as a random slope per each effect of interest, or for the resting state correlation with memory – using a linear regression and including group as a factor), to verify that the results do not stem from one group rather than the other. Plotting the data based on group in the supplementary would be useful.
7. When analyzing the behavioral results, is there a reason that the authors did not run a 3-way interaction, testing the interaction of congruency X delay X details? This interaction would confirm that indeed, the ratio of detailed vs. coarse memories changed over time, dependent on congruency, which is the argument that the authors are making. If I understood correctly, coarse and detailed memories were calculated as percentage of total trials, such that they are not fully dependent.
1. Regarding the sample size, it is unclear what is the final number of participants. Are the subjects excluded beyond the 23 mentioned? Or is the final sample of 19 participants (and 17 for the resting state connectivity)? It seems the latter from Fig. 3b, but please verify. I note that nineteen (or 23) participants seems like a low N for a complicated design with so many factors. Nineteen is also lower by about 20% of the target sample size the authors' power analysis indicated. Further, note that the study by Tomparry et al. (2017), on which the current study was based on, and the sample size was based on, did not have different types of memory, or congruent vs. incongruent trials, so it was a much simpler design (I also believe the number of items per analysis bin was larger). Previous prior knowledge/congruency studies also typically addressed only congruent vs. incongruent items, and potentially their interaction with memory. I am aware that this is an expensive and labor-intensive study, so if cost considerations also influenced the choice of the sample size, this should be stated. The issue with small sample size, aggravated by potentially small number of trials per bin (see below), is that extreme numbers might sway the results. Thus, I would strongly encourage the authors to plot the spread of participants' data in each graph. As the main interest is of within-participant differences, the authors can also plot the spread of within-participant differences for the main comparisons of interest (for example, in figure 4b, calculate the difference between congruent-within and congruent-across for each participant in each delay condition and plot that (either along the figure, or in the supplementary), and likewise for incongruent. That would be informative to evaluate the results, especially given the relatively low N in a complicated design. Additionally, please report how many participants in the final sample were from each counterbalancing group.
8. Congruency was determined based on subjects' responses (at least 2 trials of the 3). If I understand correctly, this also determined the status of the trials as congruent or incongruent for the fMRI analysis.
 - a. The authors should report the number of trials classified as congruent vs. incongruent, ideally also by memory outcome, to evaluate whether the authors had enough trials for the fMRI analyses.
 - b. Further, since participants saw each pair 3 times, and there are only two response options (related/unrelated), a participant that responds randomly, or is uncertain about the status, might inevitably have 2 responses per congruent or incongruent. It

would be informative to know how many trials had 3 consistent responses, vs. 2 responses, and whether all analyses in the paper hold (even just numerically, as power is reduced), when taking only trials that were categorized consistently during encoding.

Minor concerns:

2. Is the connectivity between hippocampus and mPFC related to representational changes in mPFC for congruent items after a long delay? The authors could examine this, across participants, by correlating the difference in connectivity (post-encoding vs. baseline), with memory for congruent items, and potentially only coarsely remembered items, or with the increase in coarsely remembered congruent items from short to long delay per participant. If so, it would imply that hippocampal-cortical interactions might mediate cortical representational changes, which would inform neuroscientific theories. Even if that result is not there, I'd encourage the authors to report it in the supplementary. It is of course a null result, but since Tomparry et al. (2017) reported a similar correlation, it would be useful to the field to know that this effect did not replicate in the current study.
3. For the background connectivity analysis, reporting a significant correlation in one condition but not in another is not sufficient. To argue that the correlation of connectivity with memory is specifically related to memory for long-delay congruent coarse memory, the authors should compare this correlation value to other correlations: short-delay congruent coarse memory, long-delay congruent detailed memory, and long-delay incongruent coarse memory. These comparisons could be done using a permutation approach replacing labels and testing for the difference between correlations or by using parametric approach to compare correlation values. The interpretation of this finding should be adjusted based on the specificity as indicated by the comparison between correlations.
4. The authors do not have a clean comparison of post-immediate encoding rest scan, which could contribute to noisier relationship with immediate memory. In both groups, another design would also include a rest scan between immediate encoding and retrieval, and would use these scans to control for resting state connectivity influences on immediate memory. I understand there are limitations of the total duration of the scan. However, this limitation of the design should be clearly mentioned in the Methods, and discussed in the Discussion.
5. In RSA correlations for scene granularity the authors mention anterior/posterior hippocampus, while the ROI selection is not mentioned in the RSA for congruency. Please edit so that this is consistent for both analyses.
6. The focus on the right hippocampus is unclear to me. I see the argument for visual memory, but previous studies addressing prior knowledge influences on the hippocampus (some have used visual stimuli as well) do not show a clear lateralization (e.g., van Kesteren et al., 2010 used bilateral hippocampus; van Buuren et al., 2014, reported bilateral clusters; Bein et al. 2014 and Brod et al., 2016, and van der Linden et al. 2017, indeed found a right hippocampal effects, but Reggev et al., 2016, and Bein et al., 2020 reported an effect in the left hippocampus). And, the Tomparry et al. (2017) paper that this study is based on used bilateral hippocampus. I'd recommend the authors to either use bilateral hippocampus, or run the analysis with both left and right hippocampus, and testing whether there is

interaction with hemisphere. In any case, the results in the left hippocampus should be reported in the supplementary. The focus on the anterior hippocampus for the connectivity analyses is clear, and justified by the anatomy (as the authors mention), and by previous findings (Brod et al., 2016; Bein et al., 2020; Tompary & Davachi, 2017; Reggev et al., 2016).

7. The motivation for the hippocampus, and hypothesizing an effect regardless of congruency seems a bit strange to me. There are theoretical frameworks hypothesizing different hippocampal involvement in processing congruent vs. incongruent information (McClelland et al., 1995; McClelland, 2013; van Kesteren et al., 2012; Gilboa et al., 2017) as well as accumulating empirical evidence, as mentioned above. It might be truer to current literature to pose that as a question – whether hippocampal involvement here is modulated by congruency or not (over time). Further, it seems from Supplementary Figure 5 that there is a difference between congruent and incongruent items in the anterior hippocampus, over long delay (though maybe not significant?). Please elaborate on that. It is interesting that the hippocampal findings are potentially not modulated by congruency in this experiment, and I'd consider discussing that point in the Discussion and relating it better to previous findings.
8. Behavior: The authors report calculating the percentage correct per participant (per different conditions), then using mixed-effects models. I'm confused about the usage of mixed-effects models here – these are usually used for trial-level analysis. What are the multiple levels? Is it just a linear model with an intercept per participant to account for the fact that this is a within-participant design? Please explain. If want to use a mixed-effect analysis, maybe a better way would be to use single trial responses and run a logistic regression, or a multinomial logistic regression (I do not believe mixed-effects are necessarily needed for the behavioral analysis, just please clarify)?
9. I think that some of the numbers of the references might be misaligned? I haven't reviewed all of them, but the references in the Introduction might not correspond to the content. Please check.
10. L. 70: The sentence “Although paradigms measuring integration as a function of shared arbitrary features can speak to the role of overlap in linking episodic memories, we argue that the complex and abstracted real-world knowledge that comprise schemas likely affect representation in the brain differently.” is confusing. That is, it makes sense that schemas, learned over a lifetime, having meaningful semantic context, are different than arbitrary overlap – but, if the authors should specify how they think schemas are different. Currently, this remains unclear.
9. L. 98: I'm confused about point (3) at the end of the Introduction. The authors speak about integration with existing schemas, but the analysis did not measure that. The analysis is about integration between learned items, based on context and congruency, but I believe we don't know whether these learned items are integrated with prior existing schemas. To clarify my point, the authors did not measure, e.g., the representation of the 'beach' schema before learning and looked at how items are represented w.r.t this beach representation. This would be indeed very tricky to do, and I'm not saying that should have been done, but please clarify what you mean by point 3, or edit.
11. Regarding the control for univariate analysis, it is preferable to include the trial-level univariate activation as an additional predicting variable in the linear-mixed effect model of

the RSA and see that the effects of interest still hold (which they should). I would recommend the author to incorporate this control.

We would like to thank the reviewers for taking the time to read and comment on our manuscript. The reviewers have raised some insightful and interesting questions, some of which required additional analyses, which we address below.

Reviewer #1 (Remarks to the Author):

Summary:

In this manuscript, Audrain and colleagues examine how semantic congruency and contextual overlap influence associative memory across time, while also exploring how such memories are represented in the hippocampus and mPFC. Participants were scanned while they encoded object-context pairs that were either congruent or incongruent with semantic knowledge. They then retrieved these associations either after a short (10 minute) or long (3 day) delay. The granularity of retrieved memories was measured by testing whether participants remembered only the general context paired with each object (beach vs. kitchen) or the specific scene image that was studied (e.g., big beach vs. small beach). Behaviorally, the authors found that when object-context associations were semantically congruent (e.g., beach—seahorse), participants were more likely to remember them at a coarse level of granularity after a long vs. short delay. This same effect was not observed for incongruent associations, taken to reflect that only memories consistent with existing knowledge schemas had grown increasingly abstract over time. Several neuroimaging results were highlighted: (1) First, increased coupling between anterior hippocampus and mPFC after encoding was associated with a greater percentage of coarsely-remembered congruent associations after a delay. (2) In mPFC, retrieved memory representations were more similar between pairs that shared the same context (vs. different contexts) — but only when object-context associations were congruent with semantic knowledge. (3) After a delay, memory representations in the posterior hippocampus were more similar when they shared the same specific scene, whereas the anterior hippocampus clustered representations according to their shared context, but did not differentiate between specific scenes. These imaging results collectively demonstrate that prior knowledge and contextual overlap influence how memories come to be represented in the brain: whereas the mPFC groups memories primarily based on their accordance with an existing schema, the hippocampus organizes memories based on contextual/representational overlap at multiple levels of granularity.

The results extend previous work to examine how overlapping long term memory associations are influenced by the congruency of encoded information. Examination of how congruent compared to incongruent associations are encoded and perhaps differentially consolidated is of critical importance in our understanding of memory transformations. These results will add to the growing literature in this area. There are, however, clarifications of the bases for some conclusions made in the paper.

Major concerns:

1. The authors frequently reference how their results provide support for “accelerated” consolidation. Although these findings clearly show that schematic knowledge and/or contextual overlap influence the transformation/organization of memories over time, it’s less evident how they establish acceleration per se. The most compelling result in support of this idea seems to be that the percentage of coarse incongruent memories was marginally greater during the long vs. the short delay, but not to the same degree as congruent memories (suggesting that these incongruent pairs might eventually be abstracted in the same way that congruent pairs were). But it’s still unclear how the mPFC similarity results speak directly to this idea, as the discussion suggests. (Unless perhaps we compare the present results with those of Tomparry & Davachi, 2017, which do show evidence of integration between incongruent associations that share the same context, but only after a week-long delay. While this paper is mentioned in the manuscript, it isn’t described as critical for the idea of acceleration?) The authors should either explain the rationale more clearly behind this conclusion, or revise their claims about acceleration in the Intro/Discussion.

Thank you for pointing this out. We have shown that memories congruent with prior knowledge become qualitatively coarser over 3 days than incongruent memories, that post-encoding connectivity (a marker of consolidation) between the hippocampus and mPFC is associated with coarse congruent memories on an individual subject basis across the 3 days, and that representations of congruent memories become more similar to each other over the same time period. We did not observe the same evidence of consolidation in the incongruent condition across the 3 days for any of these three measures. These findings are to be expected if schemas accelerate consolidation.

To your point, we did not measure a third longer timepoint that would be required to show that incongruent memories “catch up” or show similar signs of neocortical integration over a longer time period. The Tompary and Davachi 2017 data, which our design was modelled upon, suggest that integration does occur for incongruent pairs in the mPFC over a longer time frame (1 week). To our knowledge, no study to date has shown that incongruent memories “catch up” to congruent within the same study, in animals or in humans. Even in the original Science paper describing accelerated consolidation in rodents by Tse et al. (2007), the authors relied on previous work as evidence that in the absence of schemas, consolidation takes longer than what they observed.

With this in mind, we believe that an accelerated consolidation interpretation is not unreasonable given that it is grounded in past research, and given the observed behavioural and neural differences between congruent and incongruent stimuli across our three measures of consolidation employed in this study. To address the reviewer’s concerns, we have changed the following: 1) We have adjusted our wording throughout to reflect that our data is congruent with an accelerated consolidation view while tempering stronger claims that suggest this was tested directly in our study, 2) have explicitly stated that further work is needed to show that incongruent memories also become integrated at a later timepoint within the same study (page 24), and 3) we thoroughly discuss alternative interpretations on (page 22-25 of the discussion).

2. The rationale behind why the correlation between ant. hipp—mPFC connectivity and coarse memory performance *must* reflect a consolidation mechanism (rather than something like memory specificity) is somewhat confusing. It’s argued that this conclusion stems from the fact that there was no such correlation between post-encoding connectivity and coarse memory at the short delay. However, the post-encoding rest periods in this study followed the encoding of pairs that were tested after the long delay — not the short delay. Particularly for participants in Group B (where rest scans were collected on an entirely different day than when the short delay pairs were studied), this conclusion feels unwarranted, as rest periods could not feasibly have captured consolidation-related activity for these short delay pairs.

Correlating memory across the short delay with post-encoding connectivity collected after encoding for the long delay was simply a way to test that the observed relationship between connectivity and memory did not merely reflect memory granularity. It could be the case that increased connectivity between the anterior hippocampus and mPFC is associated with coarse memory regardless of consolidation processes. If this were the case, connectivity should also correlate with coarse memory across the short delay, even though connectivity was not measured after the short delay. In other words, the finding should generalize regardless of when the data were collected, if the post-encoding period is not important (as would be the case if this was not a consolidation phenomenon). However, we acknowledge that this is not a perfect test. We have expanded upon the reasoning as to why we ran this test on page 10, in addition to tempering our wording regarding the conclusions we can draw from it.

3. In the mPFC similarity analyses, it sounds like the pairwise trial comparisons that go into the computation of the “incongruent across-context” bin will include some trials in which there is a semantic relationship between the scene in one pair and the object in another (e.g., “beach—spoon” vs. “kitchen—seahorse”). Although this would only be true of a subset of these pairs, could it contribute at all to the lack of a difference between within vs. across-context associations in the incongruent bin? (perhaps because some of the across-context comparisons actually include pairs with semantic overlap, thereby boosting similarity?)

Thank you for this interesting idea. To test this, we used the incongruent arbitrary object trials on their own (i.e. excluding any context-related kitchen or beach objects), and correlated mPFC patterns according to whether the arbitrary object had been paired with the same incongruent context versus with opposing incongruent contexts. The result did not change substantively. There was a main effect of delay ($F(1,6228)=47.95, p<0.0001$) but no effect of context ($F(1,6228)=0.19, p=0.67$), and no delay by context interaction ($F(1,6228)=2.08, p=0.15$). Thus, it does not appear to be the case that the lack of a difference within vs across contexts at the long delay for incongruent pairs was driven by inflated correlations due to semantic overlap in the across-context condition. We now report this analysis in the supplementary material (Supplementary Method 7).

4. All pattern analyses seem to have been conducted on trials with accurate performance (whether coarse or fine-grained). Assuming there are enough trials to examine (at least within the long delay timepoint), has it been verified that these effects do not emerge when memory is unsuccessful?

We now include plots in the supplementary material (Supplementary Method 10) of the mPFC and hippocampal RSA contrasts using forgotten trials. The effects of interest do not emerge across the long delay in any ROI. The only analysis that is replicated in forgotten trials is the finding of greater within vs across context correlations in congruent trials in the mPFC. As these results appear to be driven by a large difference between within vs across context congruent correlations at the short delay, which was the condition with the fewest forgotten trials ($M=4.48$ trials forgotten, $SD=4.44$), they may be spurious.

Minor concerns:

5. Because the data in Figure 4b are meant to summarize results obtained from 3 separate models, it would be helpful if the authors could direct attention to which particular bars on the graph reflect the important comparisons in each model. For example, the first model described in the text tells us that the two “congruent within” bars (long vs. short delay) show a greater difference than the two “incongruent within” bars. As-is, the text is somewhat challenging to connect to the figure.

Thank you for pointing this out, we now include direct references to the bars in question throughout the statistical analysis paragraphs for Figure 4b, on pages 13-15.

6. Why might similarity in mPFC and the hippocampus between pairs in all conditions be going up over time, even when there is no content overlap? The authors could offer some speculative explanations in the discussion: for example, maybe the fact that all associations were studied in the same general context (e.g., an fMRI experiment) led them to become more similar after a delay (similar to what has been previously linked to anterior hipp-mPFC connectivity in Cowan et al., 2021).

Thank you for this interesting interpretation. We considered the non-specific increase in pattern similarity in the incongruent condition to be a bit of a mystery as well, and had not considered that it could be due to a shared broader context at encoding. We have added this thought to the discussion on pages 22-23.

7. The distinction between contextual similarity vs. semantic/schematic similarity feels at times less clear-cut than the authors suggest. Although the relationship between a context (beach) and an item (seahorse) is certainly more abstract/high-level than the relationship between two different beaches, both rely on our ability to generalize across various exemplars linked to the same underlying semantic construct (beaches). Is the key distinction in this study that only contextual overlap involves any notable perceptual similarity between pairs? Clarifying the ideas here might help shed light on why the hippocampus is apparently agnostic to semantic congruency in the present study.

We appreciate the reviewer raising this point. We have gone through the manuscript to ensure consistent use of terminology referring to contexts (kitchen/beach), scenes (which specific kitchen or beach scenes) and congruency (the relationship between the object and context).

In addition, we now articulate more explicitly where and why our findings support an effect of congruency (cases where the relationship between the object and context drives the effect rather than overlapping perceptual features due to the background image), versus an effect of perceptual similarity (cases where

the effect is driven by perceptual differences between the scenes within the same schematic context or category).

We agree that context and schematic congruency must both be inherently linked to the same construct. One must first know what a beach is and what tends to be in a beach context in order to know what is congruent or incongruent with it. However, we can learn about the relative contribution of congruency beyond contextual/perceptual similarity by examining the effects of object-context congruency when context/perceptual similarity is matched between conditions. Our data suggest that in the mPFC, perceptual similarity is not the exclusive basis for integration. We observed evidence of integration of congruent object-context pairs even when contexts were matched (i.e. we compared not just within vs across context correlations as in Tompary and Davachi, but also within context correlations to other within context correlations where the difference was if the item was congruent with the context or not, keeping the context the same across conditions). This suggests that integration is facilitated by item-context congruency beyond context-context overlap or perceptual overlap.

In the hippocampus, we observed a different pattern. In the posterior hippocampus, the specific scene with which the item was paired with (beach1 vs beach2) dictated representation, while in the anterior hippocampus, the general context was more important (kitchens vs beaches). In other words, perceptual similarity/overlap mattered in the posterior hippocampus. We did not see evidence of integration for congruent pairs beyond incongruent when background scenes were matched (at least for trials remembered in detail).

Reviewer #2

The manuscript addresses an important and timely question, namely how prior knowledge influences memory transformation over time. This is an interesting paper, linking together exciting psychological and neuroscientific questions and using an elegant and suitable task and fMRI methods to answer these questions. Further, conducting such a complicated multi-session fMRI study it is a herculean effort which should be recognized. The Introduction is beautifully written - clear and concise, and the fMRI methods are largely solid. However, I do have concerns about some aspects of the design and the logic of some of the analyses. Further, the specificity of the results should be clarified to evaluate their meaning and significance. I detail these concerns below.

Major concerns:

1. I am confused about some choices made for the RSA analyses, and how to interpret them considering the behavioral effects. The main behavioral result emphasized by the authors is the increase in coarse memory over time, specifically for congruent pairs. The authors interpret that as reflecting schematization of congruent items with time and the loss of details. The resting state connectivity analyses indeed speaks to that result. However, the RSA analysis addressing congruency collapses across detailed and coarse memories. And, collapsed across both levels of memory, the authors interpret this result as integration/schematization as well. The analyses in the hippocampus are only for detailed memories. It is unclear to me what motivates these choices for the RSA analyses, or how the RSA analyses inform us about the behavioral effect.

The difference in analysis choice lies in what 'coarse' is in terms of quality of memory versus representation. One of the aims of the study was to understand how quality of memory changes with consolidation in the context of a schema. A coarse quality of memory (the experience of memory, the behavioural part of the experiment) does not include details, which is why in the behavioural measure we separate coarse and detailed memories. For the connectivity analysis, we are focusing on a correlation with this behavioural measure, based on a specific a priori hypothesis regarding potential neural mechanisms of consolidation.

For the RSA analysis it is less clear that there should be a difference in the neural representation of coarse **features** of coarse and detailed memories. This is because you cannot have a detailed memory representation that does not contain coarse features – they are not independent. For example, two very similar beaches share coarse features such as the ocean, palm trees, and sand, even though there are perceptual differences between the scenes in terms of colour, spatial information, size of these attributes, etc. Since coarse features should be present in the neural representation of both coarse and detailed memories, collapsing across coarse and detailed memories is sensible in the RSA analysis when one is looking for neural representation of coarse features. Further, if the goal is to examine the neural representation of coarse features of the memory, comparing neural representations for coarse memories to the neural representation of detailed memories would be a problematic contrast given that detailed memories also contain coarse features. In addition, as described below, we don't have enough trials in each coarse condition once broken down according to congruency and delay, to explore coarse memories on their own (<10 trials on average per person).

Note that while we cannot isolate coarse neural representations by comparing coarse and detailed memories, we can isolate the representation of detail. This is because coarse memories by their definition, do not contain unique details. Thus, when we search for neural representation of detailed memories, it makes sense to isolate the correlations to that between only detailed memories.

In other words, the decision as to which trials were included in the RSA analysis depended on which features of neural representation we were interested in. We have made explicit that phenomenological coarse quality of memory does not equate to coarse features of memory examined in the RSA analysis on page 41 of the manuscript, and have clarified the relationship between the behavioural data and RSA analysis on page 23-24 in the discussion

2. If integration of the neural representations of congruent items lead to forgetting of details, should we observe higher mPFC similarity only for coarse memories, but not detailed memory? Or do the authors predict that neural integration should be only based on congruency, regardless of the details in memory (and if so why, given the behavior)?

We would have intuited the former, but our data suggests the latter. For the within-context correlation analysis in the mPFC, when we restrict the analysis to just detailed trials (Supplementary Method 8), the interaction between congruency and delay survives ($F(1,7952)=4.25$, $p=0.039$), suggesting that coarse features of even detailed congruent trials become more similar to each other than incongruent in the mPFC over time. When we re-ran the within vs across context correlation analysis for congruent trials, we found there was no longer a main effect of context ($F(1,9910)=1.01$, $p=0.32$), but there was a marginal interaction between context and delay ($F(1,9910)=3.48$, $p=0.062$) that appears to be driven by greater within than across context integration at the long delay. Thus, there may be some context-specific integration happening for detailed congruent memories as well. There was no effect of context for incongruent trials ($F(1,6262)=0.46$, $p=0.50$), and no context by delay interaction ($F(1,6262)=0.86$, $p=0.35$).

It therefore seems as though neural integration in the mPFC proceeds based on congruency rather than granularity. In contrast, our hippocampal data suggests that memory specificity is supported by the hippocampus. Perhaps loss of detail provided by the hippocampus is also an important factor in coarsening of memory, given the theory that both gist and detailed memory traces can coexist and flexibly interact (trace transformation theory). I.e. it may not be a perfect tradeoff such that as the mPFC integrates the hippocampus becomes disengaged.

Regardless, we have not shown a direct link between coarsening of quality of memory and pattern similarity in the mPFC, to your point, but we have shown that congruency effects both quality of memory and integration. We have made this explicit in the discussion on pages 23-24.

3. Somewhat relatedly, while the authors refer to their results in terms of memory or mnemonic representations, the analyses do not address memory, in the sense that there is no comparison between different levels of memory (i.e., remembered associations vs. forgotten, or detailed vs. coarse memories). Thus, it is unclear to what extent the results reported in the paper reflect participants' memory. I am not sure whether the authors have enough items to run a meaningful analysis (see below as well), but it might be that at least for congruent-long delay, which is the main interest of the mPFC analysis and a focus of the study, there might be enough to run a meaningful analysis. Likewise, for the hippocampus, where there is no difference based on congruency, the authors might be able to collapse across congruent and incongruent items, to have enough items to compare detailed and coarse memories. Especially in the posterior hippocampus, where the argument is about detailed memories, we should expect the effect in the delayed condition to be specific to items remembered in detail, but not to coarsely remembered items. If the result reported in the posterior hippocampus appears in both detailed and coarse memories, I believe the interpretation should be different, as this similarity does not correlate with detailed memories. The authors should provide such comparisons between memory levels, to the best of their ability within the constraints of the design. If this cannot happen, please refrain from relating to the findings as necessarily reflecting memory, as we do not know how they differ by memory status. Importantly, even if unrelated to memory, at least in the mPFC the results still reflect transformation of congruent vs. incongruent information over time (and in the hippocampus, transformation of information learned within same context over time, regardless of congruency). This is interesting and novel by itself, although we just might not learn from this study how these transformations relate to participants' memory for this information.

Thank you for this point. While we have reason to not break up our trials into detailed and coarse for the pattern similarity (described above), we did contrast different types of representations based on congruency and contextual overlap, which should capture different dimensions of memory in a similar way as you propose. As we observed change in pattern representation along both of these dimensions over time, it is unclear how such findings could be explained by differences in attention or perception between conditions. In addition, we now include an analysis on forgotten trials in the supplementary material. Comparing correlations derived from forgotten trials directly to remembered can lead to findings that are difficult to interpret, especially in the hippocampus where pattern separation and differentiation is thought to occur. For example, pattern similarity around zero in forgotten trials may reflect absence of memory, but in remembered trials could reflect pattern separation of remembered trials. We therefore ran the analysis on the forgotten trials separately, where the absence of the effects observed in remembered trials should be evidence that effects observed in remembered trials reflect memory. The effects of interest do not emerge across the long delay in any ROI. The only analysis that is replicated in forgotten trials is the finding of greater within vs across context correlations in congruent trials in the mPFC – but these results appear to be driven by a large difference between within vs across context congruent correlations at the short delay, which was the condition with the fewest forgotten trials ($M=4.48$ trials forgotten, $SD=4.44$), making it difficult to draw strong conclusions. However, it is clear that the effects of interest that emerge across the long delay are not present for any contrast in the forgotten trials. We now include plots in the supplementary material (Supplementary Method 10) of the mPFC and hippocampal RSA contrasts using forgotten trials.

To address your point about the posterior hippocampus specifically, we re-ran the congruency x scene x delay analysis in the posterior hippocampus, this time collapsing across coarse and detailed memories (as done in the mPFC). Once we added in the coarse trials, the effect of scene disappeared. This suggests that the emergence of representational scene specificity in the posterior hippocampus is specific to detailed memories. This figure has been added to the supplementary material in Supplementary Method 12.

- If I understand correctly, the authors interpret the increase in neural similarity as integration or schematization of congruent items (which is potentially reflected in forgetting of details, see also comment 1). Another interpretation could be that similarity merely reflects forgetting of details. Since there is an increase in coarse memory across the delay in congruent items, that could potentially lead to increased similarity only in congruent items. It seems to me like comparing specific levels of memory could alleviate this concern as well: if similarity only reflects loss of details, congruent and incongruent items remembered with the same level of specificity (coarse, or detailed) should not show a difference in similarity. If, however, similarity is driven by

congruency, congruent items should be more similar even when comparing similar levels of memory for details.

As mentioned in point 2 above, when we re-ran the mPFC analysis restricting the trials to detailed memory trials only, we continue to observe an interaction between congruency and delay, such that even detailed memories appear to become more similar to each other in the congruent condition than the incongruent. This suggests that integration in the mPFC occurs according to congruency and does not necessarily reflect loss of detail. We have added this analysis to the supplemental material (Supplementary Method 8) and now reference this in the discussion on pages 23-24.

5. One potential confound in the current study, at least for connectivity and the mPFC results, is participants' responses. Congruent items were responded with 'related', whereas incongruent with 'unrelated'. I understand this choice from the perspective of the congruency paradigm because many previous studies have used the same task (though note that some previous studies have equated responses). However, previous studies that have used related/unrelated responses, typically compared between different levels of memory (within the same response), which alleviate this concern. Here, it is possible that integration in mPFC, as well as functional connectivity with the hippocampus, is a result of integrative encoding promoted by considering items as 'related', rather than by congruency. This possibility is consistent with the lack of a replication of Tomparry et al., 2017. In their study, integrative encoding was required by the task, and they observed integration even for items that are incongruent (across a longer delay, as the authors note). Naturally, we cannot know the source of the discrepancy based on these two studies alone. To be clear, I am also not concerned by the lack of replication itself. To answer this concern as well, showing a difference between detailed and coarse congruent memories would be most convincing. Otherwise, the authors should consider in the Discussion that their results might be driven by the nature of the task, rather than congruency.

We conjecture that both integrative encoding and schematic congruency are two different but complimentary ways of exploring neural integration. We do not agree that one confounds the other, rather, we propose that they are different approaches to how integration may be achieved. In the real world, sometimes we effortfully try to make associations between elements that are not related, and sometimes we appreciate the relationship between two elements that already are. One might speculate that at the neural level, making decisions about congruency leverages those integrative processes more efficiently, as there is a scaffold available.

For example, schemas need to be built in the first place, or modified, which is likely the result of integrative encoding. As such learning is presumably more effortful and cannot leverage pre-existing associations, we hypothesize that this form of integration should take longer. To illustrate this point, in our previous work using a continuous recognition paradigm where congruency of the object and background scene was incidental (hence requiring no explicit congruency judgement; McAndrews et al., 2016), we found evidence of repetition suppression in the hippocampus only for congruent trials. For incongruent trials, we saw repetition enhancement. This pattern of findings is consistent with the idea that congruent object-scene pairs are more rapidly bound than incongruent, which requires prolonged and more effortful binding.

We have now added a paragraph to the discussion on pages 24-25 regarding differences in the nature of the encoding task between the two studies as a potential reason for discrepant findings, and discuss our ideas on integrative encoding versus congruency.

6. All the analyses collapse across both kitchen and beach scenes. The authors should include that as a factor in their analyses (for example, by including category in their statistical models) and see that the results hold when considering the different scenes, and do not stem from one scene only. It would also be useful to plot the results by category of scenes in the supplementary to.

We didn't add scene category to the models because the RSA models become singular once you compare within vs across context correlations (across context correlations always include Kitchen-Beach correlations only). Although examining pattern similarity based on category is an interesting question, we decided not to break the within-context correlations down into kitchen and beaches in the present study for the following reasons. 1) We are interested in pattern similarity based on memory specificity but not based on category per se. The benefit of having more than one category is that it ensures that we were not simply observing category effects, such as those that might arise by visual differences between beaches and kitchens, rather than the memory specificity effects we are interested in. 2) As we are not interested in differences between kitchens and beaches (beyond assessing that they are represented differently, as revealed by across-context correlations), and have no hypothesis regarding within-context differences between the two categories, we would not know how to interpret differences if we found them. 3) separating the analyses into kitchens and beaches would cut our power in half.

7. On that note, the control analysis presented in Supplementary Figure 6 is very important and a wonderful addition to the paper. I was also worried that since most of the items in the experiment were from either kitchen or beach, it could be that time-dependent integration in mPFC is promoted for the most prevalent or salient categories. This, however, is clearly not the case. It is interesting that the arbitrary objects also lead to integration (like congruent objects). This replicates Tompary et al. (2017), but also suggests that some interaction between semantic category and incongruency influences consolidation and integration, which is very interesting. The authors might want to address this in the Discussion.

Thank you for pointing this out. It's certainly an interesting finding, and like many unanticipated findings, could generate new hypotheses. We contend that it doesn't bolster the main thread of the discussion, so have left it in supplementary material for the interested reader in order to ensure clarity in the main manuscript.

8. For all the results reported in the manuscript – the authors should consider group as a factor in the analysis (e.g., as a random slope per each effect of interest, or for the resting state correlation with memory – using a linear regression and including group as a factor), to verify that the results do not stem from one group rather than the other. Plotting the data based on group in the supplementary would be useful.

We have now included random slopes for counterbalancing group in all behavioural and RSA statistical models. None of the results changed in terms of significance. The statistics have been updated throughout the manuscript.

We did not model counterbalancing group for the connectivity correlation analyses. This is because the participants who were excluded from the connectivity analysis were mostly from one counterbalancing group, leaving only 4 participants in one of the groups which would preclude an accurate estimation of variance in this group. Instead, we provide a plot in supplemental material (Supplementary Figure 13) with subjects colored in according to counterbalancing condition. The 4 subjects that underwent testing across the short delay followed by the long delay are not notably distinct.

9. When analyzing the behavioral results, is there a reason that the authors did not run a 3- way interaction, testing the interaction of congruency X delay X details? This interaction would confirm that indeed, the ratio of detailed vs. coarse memories changed over time, dependent on congruency, which is the argument that the authors are making. If I understood correctly, coarse and detailed memories were calculated as percentage of total trials, such that they are not fully dependent.

We ran the analyses separately because the two conditions are not fully independent. This is because participants could not have a detailed memory (context question correct + scene question correct) without answering the context question correctly (used to delineate coarse trials), and having a detailed memory necessitates remembering the coarse features.

10. Regarding the sample size, it is unclear what is the final number of participants. Are the subjects excluded beyond the 23 mentioned? Or is the final sample of 19 participants (and 17 for the resting state connectivity)? It seems the latter from Fig. 3b, but please verify. I note that nineteen (or 23) participants seems like a low N for a complicated design with so many factors. Nineteen is also lower by about 20% of the target sample size the authors' power analysis indicated. Further, note that the study by Tomparry et al. (2017), on which the current study was based on, and the sample size was based on, did not have different types of memory, or congruent vs. incongruent trials, so it was a much simpler design (I also believe the number of items per analysis bin was larger). Previous prior knowledge/congruency studies also typically addressed only congruent vs. incongruent items, and potentially their interaction with memory. I am aware that this is an expensive and labor-intensive study, so if cost considerations also influenced the choice of the sample size, this should be stated.

For the behavioural and RSA analyses, there were 23 participants at the short delay and 19 at the long delay. There were 17 participants in the resting state connectivity portion. We have now made this explicit in the manuscript on page 28.

It is true that our design was more complicated than Tomparry and Davachi's, but theirs was the closest study to what we hoped to achieve and the best estimate of effect sizes in that regard. We did lower the effect size extracted from their paper for our power analysis, in order to give us a more realistic power estimate. As we are sure the reviewer appreciates, it is often difficult to compute accurate power estimates for neuroimaging data, especially when using novel designs or contrasts as in the present work. Further, there are more factors that contribute to power than are typically considered in such power

analyses, such as number of trials, imaging sequence, signal-to-noise ratio, various imaging analysis and modelling choices, etc., which further complicates the picture of what is the true power in our study.

Regardless, it is true that we did not achieve the sample size we originally planned for because we ended up having to exclude more datapoints than anticipated, and the pandemic also precluded further data collection. However, our decision to model the correlations between trials rather than average correlations for each trial provides us with many, many datapoints for the RSA analysis, greatly increasing statistical power.

11. The issue with small sample size, aggravated by potentially small number of trials per bin (see below), is that extreme numbers might sway the results. Thus, I would strongly encourage the authors to plot the spread of participants' data in each graph. As the main interest is of within-participant differences, the authors can also plot the spread of within-participant differences for the main comparisons of interest (for example, in figure 4b, calculate the difference between congruent-within and congruent-across for each participant in each delay condition and plot that (either along the figure, or in the supplementary), and likewise for incongruent. That would be informative to evaluate the results, especially given the relatively low N in a complicated design.

Thank you for this point. We remind the reviewer here that we directly modelled the pairwise correlations between trials. As there are many different correlations per trial, and as the number of correlations varies depending on how many trials the participant got correct for each condition, calculating a measure of difference is not straightforward. Below you will find the plots of the RSA correlations overlaid on the bar plots for the main RSA figures/analyses in the manuscript (Figures 4 and 5). As you can see, there are few outliers, and there don't appear to be aberrant datapoints driving the effects that comprise the main effects of interest in the manuscript.

12. Additionally, please report how many participants in the final sample were from each counterbalancing group.

After subject exclusion, 10 subjects completed the short delay followed by the long delay, and the remaining 13 completed the long delay followed by the short delay. For the connectivity analysis 4 subjects completed short delay followed by long delay, and the remaining 13 completed the long delay followed by the short delay. We now report this on page 29.

13. Congruency was determined based on subjects' responses (at least 2 trials of the 3). If I understand correctly, this also determined the status of the trials as congruent or incongruent for the fMRI analysis. The authors should report the number of trials classified as congruent vs. incongruent, ideally also by memory outcome, to evaluate whether the authors had enough trials for the fMRI analyses.

On average, there was roughly a 40/40 split between items judged as congruent or incongruent at encoding, at each delay (At the short delay judgements for congruent: $M=40.30$, $SD=1.52$; incongruent: $M=39.70$, $SD=1.52$. At the long delay judgements for congruent: $M=40.84$, $SD=1.57$; incongruent: $M=39.05$, $SD=1.61$). We now report this on page 34.

Below we detail the number of trials per each condition at each delay. The reviewer will note that the lowest number of trials are in the coarse memory condition at the short delay, as well as for coarse incongruent pairs at the long delay. Experiments assessing quality of memory such as this are inherently difficult to control the number of trials in each condition, as detailed and coarse memories tend to be somewhat inversely related. We mitigate the effect of low trial numbers in the coarse conditions by 1) collapsing coarse and fine trials for the RSA analysis, as coarse features should be present in both types of memories, and 2) accounting for differences in the number of trials in each condition by reporting and plotting the estimated marginal means of our models, which are unbiased by imbalances in the data (i.e. estimated marginal means estimate what marginal means should be had there been equal trial numbers in each condition).

Average number of trials at the short delay for each condition:

Fine related: $M=28.56$, $SD=7.32$

Fine unrelated: $M=24.09$, $SD=9.17$

Coarse related: $M=7.26$, $SD=4.43$

Coarse unrelated: $M=6.47$, $SD=4.23$

Average number of trials at long delay for each condition:

Fine related: $M=19.16$, $SD=6.60$

Fine unrelated: $M=10.58$, $SD=3.10$

Coarse related: $M=12.79$, $SD=4.43$

Coarse unrelated: $M=7.47$, $SD=2.91$

14. Further, since participants saw each pair 3 times, and there are only two response options (related/unrelated), a participant that responds randomly, or is uncertain about the status, might inevitably have 2 responses per congruent or incongruent. It would be informative to know how many trials had 3 consistent responses, vs. 2 responses, and whether all analyses in the paper hold (even just numerically, as power is reduced), when taking only trials that were categorized consistently during encoding.

On average participants answered 81.58% of trials consistently (same response for all 3 encoding runs), which is well above chance and suggests that participants were not simply randomly responding (we have added this information to page 34 of the manuscript). The lack of consistency in the remaining trials is possibly due to the fact that the encoding task proceeded quite quickly (they had 1 second to respond related or unrelated). This interpretation is supported by the fact that 25% of 'inconsistent judgements' on average were actually due to missed button responses rather than a discrepant response (i.e. two 'related' or 'unrelated' responses and one missing response). As participants did not appear to be randomly responding, and given the quick nature of the encoding task, it seems reasonable to assume that 2/3 responses reflect what the participants perceived as related/unrelated.

Minor concerns:

15. Is the connectivity between hippocampus and mPFC related to representational changes in mPFC for congruent items after a long delay? The authors could examine this, across participants, by correlating the difference in connectivity (post-encoding vs. baseline), with memory for congruent items, and potentially only coarsely remembered items, or with the increase in coarsely remembered congruent items from short to long delay per participant. If so, it would imply that hippocampal-cortical interactions might mediate cortical representational changes, which would inform neuroscientific theories. Even if that result is not there, I'd encourage the authors to report it in the supplementary. It is of course a null result, but since Tomparry et al. (2017) reported a similar correlation, it would be useful to the field to know that this effect did not replicate in the current study.

We averaged pattern similarity correlations within and across context for congruent trials for each participant, and subtracted them to get a measure of within-across context pattern similarity for congruent trails for each participant (i.e. our main evidence of representational integration in the mPFC congruent condition). We correlated these values with post-pre encoding connectivity between the anterior hippocampus and mPFC. There was no reliable correlation ($r=-0.21$, $t(15)=0.85$, $p=0.41$). We therefore, did not observe evidence that post-encoding interaction with the anterior hippocampus was related to subsequent organization of congruent memories in the mPFC. We now report this in Supplementary Method 9.

16. For the background connectivity analysis, reporting a significant correlation in one condition but not in another is not sufficient. To argue that the correlation of connectivity with memory is specifically related to memory for long-delay congruent coarse memory, the authors should compare this correlation value to other correlations: short-delay congruent coarse memory, long-delay congruent detailed memory, and long-delay incongruent coarse memory. These comparisons could be done using a permutation approach replacing labels and testing for the difference between correlations or by using parametric approach to compare correlation values. The interpretation of this finding should be adjusted based on the specificity as indicated by the comparison between correlations.

We have now conducted parametric tests of the differences between slopes of the various conditions. When we directly compared correlations between connectivity and each memory condition across the long delay using William's test for dependent correlations, we found that the correlation between connectivity and coarse congruent memory was not reliably different from that for coarse incongruent memories ($t=0.92$, $p=0.37$), but was marginally different from that for detailed congruent memories ($t=2.03$, $p=0.06$), and was reliably different than that for detailed incongruent memories ($t=3.13$, $p=0.01$). As the correlations between post-encoding connectivity and these other memory conditions were exploratory and we did not have a priori hypotheses about them, we include this information in Supplementary Method 4 for the interested reader, alongside a plot of all of the slopes for these additional correlations (Supplementary Figure 4) so that the reader can appreciate how similar versus different the slopes are to each other on the same scale.

17. The authors do not have a clean comparison of post-immediate encoding rest scan, which could contribute to noisier relationship with immediate memory. In both groups, another design would also include a rest scan between immediate encoding and retrieval, and would use these scans to control for resting state connectivity influences on immediate memory. I understand there are limitations of the total duration of the scan. However, this limitation of the design should be clearly mentioned in the Methods, and discussed in the Discussion.

Our aim with that analysis was not to test if consolidation-related processes were also operating for the short delay data, or to control for connectivity influences on immediate memory, but instead was to test if the relationship between connectivity and coarse congruent memory was evident regardless of when the

data were collected, which would suggest the observed relationship reflected memory granularity more so than consolidation-related processes (if what we observed is truly consolidation-related, connectivity should specifically be related to memory subsequently retrieved, and not to memory at an unrelated timepoint). We have clarified this on page 10.

18. In RSA correlations for scene granularity the authors mention anterior/posterior hippocampus, while the ROI selection is not mentioned in the RSA for congruency. Please edit so that this is consistent for both analyses.

We have added a line to page 12 that makes explicit that we are testing a hypothesis (outlined in the introduction) about the mPFC for the congruency analysis, and therefore use an mPFC ROI.

19. The focus on the right hippocampus is unclear to me. I see the argument for visual memory, but previous studies addressing prior knowledge influences on the hippocampus (some have used visual stimuli as well) do not show a clear lateralization (e.g., van Kesteren et al., 2010 used bilateral hippocampus; van Buuren et al., 2014, reported bilateral clusters; Bein et al. 2014 and Brod et al., 2016, and van der Linden et al. 2017, indeed found a right hippocampal effects, but Reggev et al., 2016, and Bein et al., 2020 reported an effect in the left hippocampus). And, the Tomparly et al. (2017) paper that this study is based on used bilateral hippocampus. I'd recommend the authors to either use bilateral hippocampus, or run the analysis with both left and right hippocampus, and testing whether there is interaction with hemisphere. In any case, the results in the left hippocampus should be reported in the supplementary. The focus on the anterior hippocampus for the connectivity analyses is clear, and justified by the anatomy (as the authors mention), and by previous findings (Brod et al., 2016; Bein et al., 2020; Tomparly & Davachi, 2017; Reggev et al., 2016).

We argue that collapsing across hippocampi is not ideal given the wealth of literature showing lateralized differences in hippocampal function. Introducing the left hippocampus into the main manuscript simply warrants more comparisons with fewer degrees of freedom, potentially introducing 4 way interactions which would be difficult to interpret, especially given that we have ample reason for focusing on the right hippocampus given the visual nature of the task (as described in the manuscript). However, we now present a plot of pattern similarity in the left hippocampus for detailed memory trials in Supplementary figure 12 for the interested reader. There appears to be no effect of scene in the left hippocampus.

20. The motivation for the hippocampus, and hypothesizing an effect regardless of congruency seems a bit strange to me. There are theoretical frameworks hypothesizing different hippocampal involvement in processing congruent vs. incongruent information (McClelland et al., 1995; McClelland, 2013; van Kesteren et al., 2012; Gilboa et al., 2017) as well as accumulating empirical evidence, as mentioned above. It might be truer to current literature to pose that as a question – whether hippocampal involvement here is modulated by congruency or not (over time). Further, it seems from Supplementary Figure 5 that there is a difference between congruent and incongruent items in the anterior hippocampus, over long delay (though maybe not significant?). Please elaborate on that. It is an interesting that the hippocampal findings are potentially not modulated by congruency in this experiment, and I'd consider discussing that point in the Discussion and relating it better to previous findings.

We would expect different hippocampal involvement in processing congruent and incongruent information in terms of level of engagement, as the papers you pointed to suggest, but it is less clear to us that there should be differences in how the hippocampus fundamentally represents such information when it is engaged. One idea is that the hippocampus is “stupid”, in that it doesn't particularly care about content, and simply cares about binding together relational information such as object and context (Moscovitch, 2008). We have elaborated upon these ideas in the discussion on pages 26-27.

21. Behavior: The authors report calculating the percentage correct per participant (per different conditions), then using mixed-effects models. I'm confused about the usage of mixed-effects models here – these are usually used for trial-level analysis. What are the multiple levels? Is it just a linear model with an intercept per participant to account for the fact that this is a within-participant design? Please explain. If want to use a mixed-effect analysis, maybe a better way would be to use single trial responses and run a logistic regression, or a multinomial logistic regression (I do not believe mixed-effects are necessarily needed for the behavioral analysis, just please clarify)?

You are correct that it's just a linear model with an intercept for each participant to account for the within-subject design. A mixed effects model with a random intercept for each subject using our current design is the same thing as running a repeated measures anova, we just call it a mixed effects model for consistency and because it was run with the mixed effects model package in R. In addition, we now include random slopes for each counterbalancing group.

22. I think that some of the numbers of the references might be misaligned? I haven't reviewed all of them, but the references in the Introduction might not correspond to the content. Please check.

Thank you for catching this, they had indeed become misaligned at some point. This has now been fixed.

23. L. 70: The sentence "Although paradigms measuring integration as a function of shared arbitrary features can speak to the role of overlap in linking episodic memories, we argue that the complex and abstracted real-world knowledge that comprise schemas likely affect representation in the brain differently." is confusing. That is, it makes sense that schemas, learned over a lifetime, having meaningful semantic context, are different than arbitrary overlap – but, if the authors should specify how they think schemas are different. Currently, this remains unclear.

We simply mean to point out that schemas are developed over a lifetime and evaluating a repeated arbitrary association as a knowledge structure is a step removed from what we actually care about. Learned arbitrary associations are confined to those elements that were learned in the context of the experiment, while we think schemas have multiple other elements besides what was directly encoded in the experiment, making schemas more useful for scaffolding in new knowledge. We have made this clearer on page 4.

24. L. 98: I'm confused about point (3) at the end of the Introduction. The authors speak about integration with existing schemas, but the analysis did not measure that. The analysis is about integration between learned items, based on context and congruency, but I believe we don't know whether these learned items are integrated with prior existing schemas. To clarify my point, the authors did not measure, e.g., the representation of the 'beach' schema before learning and looked at how items are represented w.r.t this beach representation. This would be indeed very tricky to do, and I'm not saying that should have been done, but please clarify what you mean by point 3, or edit.

Thank you for this, we have not asked the question as to if the schema itself has changed due to a new element being added. We have re-worded to make it clear that we are interested in the organization and overlap of schema-congruent memories.

25. Regarding the control for univariate analysis, it is preferable to include the trial-level univariate activation as an additional predicting variable in the linear-mixed effect model of the RSA and see that the effects of interest still hold (which they should). I would recommend the author to incorporate this control.

We didn't add the trial-level univariate response to the models because we modelled the correlations directly, rather than averaging correlations for each trial. Because of this, there are more correlations

being modelled than univariate trial-wise activity. In other words, it's unclear how univariate activity could be regressed out this way, because two trials (and associated univariate activity) contribute to each correlation. We have added this consideration to the description of the univariate analysis for clarity (Supplemental Method 16).

REVIEWER COMMENTS

Reviewer #1 (Remarks to the Author):

The authors have been incredibly responsive to the concerns outlined in my original review. The clarifications they propose regarding some of the interpretational questions is satisfactory and the additional analyses provided help to address my concerns. I think these results will make a strong impact.

Reviewer #2 (Remarks to the Author):

See attached.

Dear editor,

I thank the authors for the thorough response to my comments. The authors answered most of my concerns, and the manuscript has improved. I detail the remaining concerns below, keeping the original numbering for reference. My original comment is in blue, the authors respond is in bold, and my new comments are in regular font.

3. Somewhat relatedly, while the authors refer to their results in terms of memory or mnemonic representations, the analyses do not address memory, in the sense that there is no comparison between different levels of memory (i.e., remembered associations vs. forgotten, or detailed vs. coarse memories). Thus, it is unclear to what extent the results reported in the paper reflect participants' memory. I am not sure whether the authors have enough items to run a meaningful analysis (see below as well), but it might be that at least for congruent-long delay, which is the main interest of the mPFC analysis and a focus of the study, there might be enough to run a meaningful analysis. Likewise, for the hippocampus, where there is no difference based on congruency, the authors might be able to collapse across congruent and incongruent items, to have enough items to compare detailed and coarse memories. Especially in the posterior hippocampus, where the argument is about detailed memories, we should expect the effect in the delayed condition to be specific to items remembered in detail, but not to coarsely remembered items. If the result reported in the posterior hippocampus appears in both detailed and coarse memories, I believe the interpretation should be different, as this similarity does not correlate with detailed memories. The authors should provide such comparisons between memory levels, to the best of their ability within the constraints of the design. If this cannot happen, please refrain from relating to the findings as necessarily reflecting memory, as we do not know how they differ by memory status. Importantly, even if unrelated to memory, at least in the mPFC the results still reflect transformation of congruent vs. incongruent information over time (and in the hippocampus, transformation of information learned within same context over time, regardless of congruency). This is interesting and novel by itself, although we just might not learn from this study how these transformations relate to participants' memory for this information.

Thank you for this point. While we have reason to not break up our trials into detailed and coarse for the pattern similarity (described above), we did contrast different types of representations based on congruency and contextual overlap, which should capture different dimensions of memory in a similar way as you propose. As we observed change in pattern representation along both of these dimensions over time, it is unclear how such findings could be explained by differences in attention or perception between conditions.

In addition, we now include an analysis on forgotten trials in the supplementary material. Comparing correlations derived from forgotten trials directly to remembered can lead to findings that are difficult to interpret, especially in the hippocampus where pattern separation and differentiation is thought to occur. For example, pattern similarity around zero in forgotten trials may reflect absence of memory, but in remembered trials could reflect pattern separation of remembered trials.

I agree differences over time cannot be explained by differences in attention or perception during encoding. The differences over time do speak towards how information is transformed in the brain. The addition of the forgotten trials separately is beneficial, because numerically, they differ from remembered trials. However, in the absence of showing a significant difference when directly comparing different levels of memory (e.g., remembered vs. forgotten) – the RSA effects cannot be strongly interpreted as mediating memory. In addition, I agree that patterns similarity around zero can mean multiple things. In fact, not only zero but any correlation value per se is not so informative, unless compared to another condition (because the value of a correlation can be influenced by, e.g., autocorrelation in the BOLD signal, the specific structure, etc). That is why we typically want to interpret a *difference* in correlations between two conditions, like the authors do for other analyses through the paper. The same logic applies

for this analysis: to argue that there is a difference between any one condition to another, in this case remembered and forgotten trials, it does not suffice to show an effect emerge in one condition but not in another – one should directly compare these conditions. This was done in numerous previous studies comparing remembered to forgotten trials (or different levels of memory) to argue that RSA mediates memory (specifically in RSA, e.g.: Bosch et al., 2014; Ritchey et al., 2012; Staresina et al., 2012; Wing et al., 2015; Tompary et al., 2016, 2017). Does the logic of these previous papers not apply here? If the authors find a significant difference when directly comparing remembered and forgotten trials, that would allow interpreting the results as mediating memory. Otherwise, I would recommend to include the forgotten trials (as currently is) in which the authors show no similarity, but remove any language that strongly interprets these RSA results as specific to memory (e.g., in p. 16, l. 334, line 339-340). The numeric differences are suggestive, but more research is needed to determine whether this difference is reliable, and the authors should specifically mention that. As I mentioned previously, I think the results are novel and interesting even if the link to memory is not strongly apparent in these data.

To address your point about the posterior hippocampus specifically, we re-ran the congruency x scene x delay analysis in the posterior hippocampus, this time collapsing across coarse and detailed memories (as done in the mPFC). Once we added in the coarse trials, the effect of scene disappeared. This suggests that the emergence of representational scene specificity in the posterior hippocampus is specific to detailed memories. This figure has been added to the supplementary material in Supplementary Method 12.

Unfortunately, this analysis does not show specificity. It shows that when pooled together – there is no effect, suggesting that the hippocampal effect is smaller for coarse than detailed memories. But, to argue for specificity to *detailed* memory, the authors need to show that detailed memories are significantly different than coarse memories. Given the number of trials, that might not be doable separately for congruent/incongruent, but as the results are not specific to congruency, collapsing across congruency should be doable. To foreshadow, also comparing between detailed memories and forgotten trials is not sufficient to argue specifically about detailed memory. Such a comparison would be useful to show the RSA results mediate memory. But, it will still be difficult to argue specificity to detailed memory if only compared to forgotten trials, since coarse memories might show the same and may not be significantly different than detailed memories. To argue specificity to detailed memory, the authors should directly compared detailed and coarse memory.

4. All the analyses collapse across both kitchen and beach scenes. The authors should include that as a factor in their analyses (for example, by including category in their statistical models) and see that the results hold when considering the different scenes, and do not stem from one scene only. It would also be useful to plot the results by category of scenes in the supplementary to.

We didn't add scene category to the models because the RSA models become singular once you compare within vs across context correlations (across context correlations always include Kitchen-Beach correlations only). Although examining pattern similarity based on category is an interesting question, we decided not to break the within-context correlations down into kitchen and beaches in the present study for the following reasons. 1) We are interested in pattern similarity based on memory specificity but not based on category per se. The benefit of having more than one category is that it ensures that we were not simply observing category effects, such as those that might arise by visual differences between beaches and kitchens, rather than the memory specificity effects we are interested in. 2) As we are not interested in differences between kitchens and beaches (beyond assessing that they are represented differently, as revealed by across-context correlations), and have no hypothesis regarding within-context differences between the two categories, we would not know how to interpret differences if we found them. 3) separating the analyses into kitchens and beaches would cut our power in half.

I understand the authors are not interested in category effects. I agree. I asked to report these results not to explore differences based on category, but to examine the robustness of the results across the two scenes used in the study. If the models are too complex, please simply plot the main results based on

category, so the reader could evaluate that the effects are at least numerically there for both scenes (even if not statistically significant). Having more than one scene does not ensure that one is not observing a category effect. The risk is that the results occur mostly in one scene, and that's enough to draw the average across both scenes to show an overall effect. My apologies if this point wasn't clear in my previous round of reviews.

14. Further, since participants saw each pair 3 times, and there are only two response options (related/unrelated), a participant that responds randomly, or is uncertain about the status, might inevitably have 2 responses per congruent or incongruent. It would be informative to know how many trials had 3 consistent responses, vs. 2 responses, and whether all analyses in the paper hold (even just numerically, as power is reduced), when taking only trials that were categorized consistently during encoding.

On average participants answered 81.58% of trials consistently (same response for all 3 encoding runs), which is well above chance and suggests that participants were not simply randomly responding (we have added this information to page 34 of the manuscript). The lack of consistency in the remaining trials is possibly due to the fact that the encoding task proceeded quite quickly (they had 1 second to respond related or unrelated). This interpretation is supported by the fact that 25% of 'inconsistent judgements' on average were actually due to missed button responses rather than a discrepant response (i.e. two 'related' or 'unrelated' responses and one missing response). As participants did not appear to be randomly responding, and given the quick nature of the encoding task, it seems reasonable to assume that 2/3 responses reflect what the participants perceived as related/unrelated.

I agree that it says that largely, the participants were not random. But it can still be that for these 20% of responses – they are somewhat in-between. Thus, it would still be informative to know – do the results look similar when removing the 20% of responses that participants did not consistently say they are congruent/incongruent? If not, it might be that these intermediate congruency/incongruency level might contribute a lot to the results, which might lead to a different interpretation of the data. Since the authors have the data, I believe that reporting these results would clarify the robustness of the findings. I acknowledge that this reduces power, thus even if the results are numerically similar, but not significant, it would still be useful.

25. Regarding the control for univariate analysis, it is preferable to include the trial-level univariate activation as an additional predicting variable in the linear-mixed effect model of the RSA and see that the effects of interest still hold (which they should). I would recommend the author to incorporate this control.

We didn't add the trial-level univariate response to the models because we modelled the correlations directly, rather than averaging correlations for each trial. Because of this, there are more correlations being modelled than univariate trial-wise activity. In other words, it's unclear how univariate activity could be regressed out this way, because two trials (and associated univariate activity) contribute to each correlation. We have added this consideration to the description of the univariate analysis for clarity (Supplemental Method 16).

I understand that two trials contribute to each correlation. If I'm not mistaken, that means the authors can take the univariate activity in each trial as an additional regressor (i.e., add two regressors to the model). I might miss something, but that should be straight-forward. Generally speaking, the correlation measure subtracts the mean, and therefore the results should be robust, but it's useful to provide the data given that there is a risk that univariate activation might contribute to the results (Davis et al., 2014; Dimsdale-Zucker and Ranganath, 2018).

We thank the reviewer for their continued consideration of our manuscript. Please see our response below in red.

3. Somewhat relatedly, while the authors refer to their results in terms of memory or mnemonic representations, the analyses do not address memory, in the sense that there is no comparison between different levels of memory (i.e., remembered associations vs. forgotten, or detailed vs. coarse memories). Thus, it is unclear to what extent the results reported in the paper reflect participants' memory. I am not sure whether the authors have enough items to run a meaningful analysis (see below as well), but it might be that at least for congruent-long delay, which is the main interest of the mPFC analysis and a focus of the study, there might be enough to run a meaningful analysis. Likewise, for the hippocampus, where there is no difference based on congruency, the authors might be able to collapse across congruent and incongruent items, to have enough items to compare detailed and coarse memories. Especially in the posterior hippocampus, where the argument is about detailed memories, we should expect the effect in the delayed condition to be specific to items remembered in detail, but not to coarsely remembered items. If the result reported in the posterior hippocampus appears in both detailed and coarse memories, I believe the interpretation should be different, as this similarity does not correlate with detailed memories. The authors should provide such comparisons between memory levels, to the best of their ability within the constraints of the design. If this cannot happen, please refrain from relating to the findings as necessarily reflecting memory, as we do not know how they differ by memory status. Importantly, even if unrelated to memory, at least in the mPFC the results still reflect transformation of congruent vs. incongruent information over time (and in the hippocampus, transformation of information learned within same context over time, regardless of congruency). This is interesting and novel by itself, although we just might not learn from this study how these transformations relate to participants' memory for this information.

Thank you for this point. While we have reason to not break up our trials into detailed and coarse for the pattern similarity (described above), we did contrast different types of representations based on congruency and contextual overlap, which should capture different dimensions of memory in a similar way as you propose. As we observed change in pattern representation along both of these dimensions over time, it is unclear how such findings could be explained by differences in attention or perception between conditions.

In addition, we now include an analysis on forgotten trials in the supplementary material. Comparing correlations derived from forgotten trials directly to remembered can lead to findings that are difficult to interpret, especially in the hippocampus where pattern separation and differentiation is thought to occur. For example, pattern similarity around zero in forgotten trials may reflect absence of memory, but in remembered trials could reflect pattern separation of remembered trials.

I agree differences over time cannot be explained by differences in attention or perception during encoding. The differences over time do speak towards how information is transformed in the brain. The addition of the forgotten trials separately is beneficial, because numerically, they differ from remembered trials. However, in the absence of showing a significant difference when directly comparing different levels of memory (e.g., remembered vs. forgotten) – the RSA effects cannot be strongly interpreted as mediating memory. In addition, I agree that patterns similarity around zero can mean multiple things. In fact, not only zero but any correlation value per se is not so informative, unless compared to another condition (because the value of a correlation can be influenced by, e.g., autocorrelation in the BOLD signal, the specific structure, etc). That is why we typically want to interpret a difference in correlations between two conditions, like the authors do for other analyses through the paper. The same logic applies for this analysis: to argue that there is a difference between any one condition to another, in this case remembered and forgotten trials, it does not suffice to show an effect emerge in one condition but not in another – one should directly compare these conditions. This was done in numerous previous studies comparing remembered to forgotten trials (or different levels of memory) to argue that RSA mediates memory (specifically in RSA, e.g.: Bosch et al., 2014; Ritchey et al., 2012; Staresina et al., 2012; Wing et al., 2015; Tomparry et al., 2016, 2017). Does the logic of these previous papers

not apply here? If the authors find a significant difference when directly comparing remembered and forgotten trials, that would allow interpreting the results as mediating memory. Otherwise, I would recommend to include the forgotten trials (as currently is) in which the authors show no similarity, but remove any language that strongly interprets these RSA results as specific to memory (e.g., in p. 16, l. 334, line 339-340). The numeric differences are suggestive, but more research is needed to determine whether this difference is reliable, and the authors should specifically mention that. As I mentioned previously, I think the results are novel and interesting even if the link to memory is not strongly apparent in these data.

We remind the reviewer here that participants were not re-presented with the scene, they are recalling it in the absence of the dimensions for which we analyze the data (congruency and scene/context). It is still unclear what differences along these dimensions would reflect if not memory, given the lack of bottom-up contribution along these dimensions.

Regarding the pattern similarity interpretation, we agree that observing a difference in similarity between remembered and forgotten trials would be consistent with a neural representation of a memory. However, we do not concur that the lack of a statistically significant difference in representation between remembered and forgotten trials would indicate that memory is not the key feature indexed by the RSA. We of course agree that pattern similarity is only interpretable when considered in relation to informative conditions. Our argument is that the forgotten condition is not necessarily an informative condition to compare against.

To elaborate upon our earlier example, you might expect correlations between forgotten beaches and kitchens to be around zero, because the hippocampus is not engaged or representing such information when it's forgotten, therefore you are correlating noise. However, if the hippocampus is orthogonalizing beaches and kitchens during successfully remembered trials, correlations between beaches and kitchens may also be around zero. If you were to compare patterns between forgotten and remembered in this example, you would not find a significant difference, and you might erroneously conclude that the hippocampus is not involved in memory. In this example, forgotten trials are not an informative comparison point, and a lack of a statistical difference doesn't necessarily prove that memory did not occur. In our opinion, the entire context should be considered regarding behaviour and known neuroanatomy that supports that behaviour, in addition to pattern comparisons across other dimensions (as we have done with congruency and context and time).

Regardless, we appreciate this is a point of contention, so we have compared remembered and forgotten trials for the reviewer below. We restrict our statistical analysis to the long delay given that there are too few forgotten trials in some of the conditions at the short delay and given that our effects of interest are at the long delay. We do however, plot the data across both delays for visualization. Importantly, a main effect of memory status (remembered vs forgotten) is significant for all three ROIs at the long delay, and therefore remembered and forgotten trials have different pattern similarity on average.

Posterior hippocampus

■ R:samescene ■ R:xcon ■ UR:simscene
■ R:simscene ■ UR:samescene ■ UR:xcon

	DFn	DFD	F value	P value
Memory status	1	6232.6	54.65	<0.000001
Congruency	1	6677.9	1.54	0.21
Scene	2	6707.2	2.45	0.086
Memory status x congruency	1	6236.5	0.034	0.85
Memory status x scene	2	6707	1.35	0.26
Congruency x scene	2	6702.3	0.035	0.97
Memory x congruency x scene	2	6701.4	1.65	0.19

Anterior hippocampus

	DFn	DFD	F value	P value
Memory status	1	6468.1	10.38	0.001
Congruency	1	6701.8	3.34	0.068
Scene	2	6706.2	0.88	0.41
Memory status x congruency	1	6465.9	0.31	0.58
Memory status x scene	2	6705.9	2.89	0.056
Congruency x scene	2	6701.1	0.81	0.45
Memory x congruency x scene	2	6700.4	1.13	0.32

mPFC

	DFn	DFD	F value	P value
Memory status	1	11375	102.09	<0.000001
Congruency	1	11371	15.20	0.000097
Context	1	11365	1.23	0.27
Memory status x congruency	1	11379	7.29	0.0069
Memory status x context	1	11365	0.29	0.59
Congruency x context	1	11365	4.82	0.028
Memory x congruency x context	1	11365	1.09	0.30

To address your point about the posterior hippocampus specifically, we re-ran the congruency x scene x delay analysis in the posterior hippocampus, this time collapsing across coarse and detailed memories (as done in the mPFC). Once we added in the coarse trials, the effect of scene disappeared. This suggests that the emergence of representational scene specificity in the posterior hippocampus is specific to detailed memories. This figure has been added to the supplementary material in Supplementary Method 12.

Unfortunately, this analysis does not show specificity. It shows that when pooled together – there is no effect, suggesting that the hippocampal effect is smaller for coarse than detailed memories. But, to argue for specificity to detailed memory, the authors need to show that detailed memories

are significantly different than coarse memories. Given the number of trials, that might not be doable separately for congruent/incongruent, but as the results are not specific to congruency, collapsing across congruency should be doable. To foreshadow, also comparing between detailed memories and forgotten trials is not sufficient to argue specifically about detailed memory. Such a comparison would be useful to show the RSA results mediate memory. But, it will still be difficult to argue specificity to detailed memory if only compared to forgotten trials, since coarse memories might show the same and may not be significantly different than detailed memories. To argue specificity to detailed memory, the authors should directly compare detailed and coarse memory.

There was no effect of congruency for detailed memories but there was some indication of a congruency effect in the hippocampus when we collapsed across coarse and detailed (Supplementary Figures 5 and 12), so it is possible that congruency is a moderating factor for coarse memories. However, we agree that we should refrain from claiming these results are specific to detailed memory trials and not coarse. To address this concern, we have changed our wording to reflect that the supplementary analysis suggests that representational scene specificity is smaller for coarse than detailed memories, and we temper our wording to avoid implying these results are specific to detailed memory and not coarse throughout the manuscript. We have also addressed the fact that we did not show specificity for detailed and not coarse memories on page 27 of the discussion.

4. All the analyses collapse across both kitchen and beach scenes. The authors should include that as a factor in their analyses (for example, by including category in their statistical models) and see that the results hold when considering the different scenes, and do not stem from one scene only. It would also be useful to plot the results by category of scenes in the supplementary to.

We didn't add scene category to the models because the RSA models become singular once you compare within vs across context correlations (across context correlations always include Kitchen-Beach correlations only). Although examining pattern similarity based on category is an interesting question, we decided not to break the within-context correlations down into kitchen and beaches in the present study for the following reasons. 1) We are interested in pattern similarity based on memory specificity but not based on category per se. The benefit of having more than one category is that it ensures that we were not simply observing category effects, such as those that might arise by visual differences between beaches and kitchens, rather than the memory specificity effects we are interested in. 2) As we are not interested in differences between kitchens and beaches (beyond assessing that they are represented differently, as revealed by across-context correlations), and have no hypothesis regarding within-context differences between the two categories, we would not know how to interpret differences if we found them. 3) separating the analyses into kitchens and beaches would cut our power in half.

I understand the authors are not interested in category effects. I agree. I asked to report these results not to explore differences based on category, but to examine the robustness of the results across the two scenes used in the study. If the models are too complex, please simply plot the main results based on category, so the reader could evaluate that the effects are at least numerically there for both scenes (even if not statistically significant). Having more than one scene does not ensure that one is not observing a category effect. The risk is that the results occur mostly in one scene, and that's enough to draw the average across both scenes to show an overall effect. My apologies if this point wasn't clear in my previous round of reviews.

Below we present plots where the within-context correlations reflect correlations between just beaches, and just kitchens, separately. As you can see, the hippocampal results don't appear to differ depending on category, but the mPFC results seem to be mostly driven by beaches. As these analyses cut our power by 50%, and given the other issues we identified above, these results should not be over interpreted. We present these analyses in supplementary material and acknowledge that further research is required to investigate category-specific effects on page 16.

14. Further, since participants saw each pair 3 times, and there are only two response options (related/unrelated), a participant that responds randomly, or is uncertain about the status, might inevitably have 2 responses per congruent or incongruent. It would be informative to know how many trials had 3 consistent responses, vs. 2 responses, and whether all analyses in the paper hold (even just numerically, as power is reduced), when taking only trials that were categorized consistently during encoding.

On average participants answered 81.58% of trials consistently (same response for all 3 encoding runs), which is well above chance and suggests that participants were not simply randomly responding (we have added this information to page 34 of the manuscript). The lack of consistency in the remaining trials is possibly due to the fact that the encoding task proceeded quite quickly (they had 1 second to respond related or unrelated). This interpretation is supported by the fact that 25% of 'inconsistent judgements' on average were actually due to missed button responses rather than a discrepant response (i.e. two 'related' or 'unrelated' responses and one missing response). As participants did not appear to be randomly responding, and given the quick nature of the encoding task, it seems reasonable to assume that 2/3 responses reflect what the participants perceived as related/unrelated.

I agree that it says that largely, the participants were not random. But it can still be that for these 20% of responses – they are somewhat in-between. Thus, it would still be informative to know –

do the results look similar when removing the 20% of responses that participants did not consistently say they are congruent/incongruent? If not, it might be that these intermediate congruency/incongruency level might contribute a lot to the results, which might lead to a different interpretation of the data. Since the authors have the data, I believe that reporting these results would clarify the robustness of the findings. I acknowledge that this reduces power, thus even if the results are numerically similar, but not significant, it would still be useful.

Inevitably, the level of perceived congruency will vary between trials and participants to some extent that we cannot control, regardless of the binary decision that they make. By the reviewer's reasoning, the 20% of responses that participants responded inconsistently should be injecting noise into our analysis, and removing them should only improve our results (unless the loss of power counteracts this). We therefore don't expect the proposed analysis to be particularly informative or impactful such as to warrant re-running all of our analyses without said trials.

25. Regarding the control for univariate analysis, it is preferable to include the trial-level univariate activation as an additional predicting variable in the linear-mixed effect model of the RSA and see that the effects of interest still hold (which they should). I would recommend the author to incorporate this control.

We didn't add the trial-level univariate response to the models because we modelled the correlations directly, rather than averaging correlations for each trial. Because of this, there are more correlations being modelled than univariate trial-wise activity. In other words, it's unclear how univariate activity could be regressed out this way, because two trials (and associated univariate activity) contribute to each correlation. We have added this consideration to the description of the univariate analysis for clarity (Supplemental Method 16).

I understand that two trials contribute to each correlation. If I'm not mistaken, that means the authors can take the univariate activity in each trial as an additional regressor (i.e., add two regressors to the model). I might miss something, but that should be straight-forward. Generally speaking, the correlation measure subtracts the mean, and therefore the results should be robust, but it's useful to provide the data given that there is a risk that univariate activation might contribute to the results (Davis et al., 2014; Dimsdale-Zucker and Ranganath, 2018).

As we understand it, there are multiple ways one can rule out univariate effects on RSA patterns, one of which is to simply show that univariate and multivariate results differ (the approach we took, given the complexity of our models), as acknowledged in the cited Dimsdale-Zucker and Ranganath 2018 paper. We would not anticipate that re-running all of our analyses with univariate regressors would change our results given the lack of demonstrated univariate effects and given that our trial patterns were demeaned before computing correlations. While preferences in how to best handle potential univariate confounds may differ across groups, we believe the approach we took is acceptable. Especially so when one considers that univariate effects on RSA analyses are typically only of practical concern when using Euclidean measures of distance, searchlight analyses, or when there are large differences in univariate activation between conditions (which our univariate analysis rules out).

REVIEWERS' COMMENTS

Reviewer #2 (Remarks to the Author):

The authors answered my concerns.